# Semi-automated IT-scATAC-seq profiles cell-specific chromatin accessibility in differentiation and peripheral blood populations

Wei Jin [1,2,3,5] ✉, Jingchun Ma[3,5], Li Rong[3,5], Shengshuo Huang[3], Tuo Li[4], Guoxiang Jin [2] & Zhongjun Zhou [1,2,3] ✉

Single-cell ATAC-seq (scATAC-seq) enables high-resolution mapping of chromatin accessibility but is often limited by throughput, cost, and equipment requirements. Here, we present indexed Tn5 tagmentation-based scATAC-seq (IT-scATAC-seq), a semi-automated, cost-effective, and scalable approach that leverages indexed Tn5 transposomes and a three-round barcoding strategy. This workflow prepares libraries for up to 10,000 cells in a single day, reduces the per-cell cost to approximately $0.01, and maintains high data quality. Comprehensive benchmarking demonstrates that IT-scATAC-seq achieves robust library complexity, high signal specificity, and improved cost-efficiency compared to existing methods. We apply IT-scATAC-seq to mouse embryonic stem cells, capturing chromatin remodelling during early differentiation, and to human peripheral blood mononuclear cells, resolving cell-type−specific regulatory programs. Here, we show that IT-scATAC-seq provides a robust and efficient approach for high-resolution single-cell epigenomic investigations, balancing scalability, data quality, and accessibility.

Studying gene regulation at the single-cell level is becoming increasingly important for understanding cellular heterogeneity in complex biological systems[1–3]. While single-cell transcriptomics captures gene expression dynamics, single-cell epigenomics provides insights into the regulatory mechanisms underlying these profiles[1,2,4]. Assay for Transposase-Accessible Chromatin Sequencing (ATAC-seq) is a powerful tool that maps open chromatin regions that control gene expression without needing prior knowledge of epigenetic markers or transcription factors[5]. Single-cell ATAC-seq (scATAC-seq) extends this capability, enabling chromatin accessibility profiling at the resolution of individual cells. Recent advances in scATAC-seq methods – through

microfluidics-based[6] or in-house prepared single-cell combinatorial indexing (sci)-ATAC-seq[7], plate-based scATAC-seq[8] and μATAC-seq[9], alone or integrated with other single-cell omics[10–15] – have broadened our understanding of how genetic and environmental factors shape cellular identity, cell state transitions, functional variations, and disease mechanisms, thereby expanding the scope of transcriptional regulation research.

Library quality underlies accurate interpretation of scATAC-seq data. Key technical determinants include sensitivity (the ability to detect all accessible chromatin regions, reflected in library complexity), accuracy (the correspondence between sequenced fragments and

[1]Pediatric Research Institute, Dongguan Children Hospital, Guangdong Medical University, Dongguan, China. [2]Medical Research Centre; Guangdong Cardiovascular Institute; Key Laboratory for Immune and Genetic Research of Chronic Nephropathy, Guangdong Provincial People's Hospital, Guangdong Academy of Medical Sciences, Southern Medical University, Guangzhou, China. [3]School of Biomedical Sciences, LKS Faculty of Medicine, The University of Hong Kong, Hong Kong, Hong Kong. [4]Department of Endocrinology, Changzheng Hospital, Shanghai, China. [5]These authors contributed equally: Wei Jin, Jingchun Ma, Li Rong. ✉e-mail: jinwei5@connect.hku.hk; zhongjun@hku.hk

authentic ATAC-seq signals from single cells), and specificity (the ability to discern ATAC signals specific to different cell types and states). Another important consideration is to maximise the number of cells analysed while minimising time, manual effort, and cost. However, current methods struggle to simultaneously achieve high sensitivity, accuracy, throughput, and affordability, which hinders the widespread application and development of scATAC-seq technology[8,9,16]. For example, droplet-based microfluidic systems or nanowell-based platforms, such as Bio-Rad ddSEQ[17], Fluidigm C1[6], and Takara ICELL8[9], can be expensive and require specialised equipment, limiting their use only in well-resourced settings. Although plate-based scATAC-seq is relatively simple and robust, its throughput constraints analysis to hundreds to thousands of cells and further scaling results in a disproportionate increase in labour and PCR costs[8]. Meanwhile, sci-scATAC-seq boosts throughput to an organ scale through multiple rounds of splitting and pooling[7,18–20], but it often comes at the expense of compromising library quality and demands a large amount of indexed Tn5 to be prepared. These challenges highlight the need for a more cost-effective, sensitive, and scalable solution accessible to various academic and clinical applications.

To address these limitations, we developed IT-scATAC-seq, a streamlined and semi-automated method that employs a three-round indexing strategy. This approach leverages barcoded Tn5 for the first indexing, followed by two rounds of indexed PCR to achieve easy scalability. Combining parallel bulk tagmentation with fluorescence-activated nuclei sorting (FANS), IT-scATAC-seq reduces both per-cell costs and hands-on time while maintaining robust library complexity and high signal specificity.

Benchmarking analysis showed IT-scATAC-seq yields high library complexity, low mitochondrial contamination, and strong ATAC-seq signal enrichment around transcription start sites (TSS), with more than 60% of reads in peaks (FRiP). To demonstrate its utility, we applied IT-scATAC-seq to mouse embryonic stem cells(mESCs) undergoing differentiation, showing chromatin accessibility dynamics as cells transition from naïve pluripotency. Additionally, we profiled human peripheral blood mononuclear cells (PBMCs), demonstrating the method's ability to resolve distinct immune subsets and their cell-type-specific regulatory elements. Together, these findings establish IT-scATAC-seq as a cost-effective and high-throughput technology for profiling single-cell chromatin accessibility. By eliminating the need for specialised equipment and enabling library preparation for 10,000 cells in a single day at less than $0.01 per cell, IT-scATAC-seq reduces costs while maintaining high-quality data. With its scalable and efficient workflow, this method expands the accessibility of single-cell chromatin profiling, making it adaptable to various biological and clinical research contexts.

## Results

### Benchmark of IT-scATAC-seq

We developed IT-scATAC-seq, a simple and scalable strategy to profile the single-cell chromatin accessibility using indexed Tn5 tagmentation and a three-round indexing strategy (Fig. 1a, Supplementary Fig. 1 and 2, and Supplementary Data 1). In this method, nuclei are isolated following the refined Omni-ATAC protocol[21] to minimise mitochondrial DNA contamination and then divided into multiple parts for parallel bulk transposition reactions with in-house purified and assembled indexed Tn5 complexes (number of reactions = $N$) (Supplementary Fig. 1 and 2). The transposed nuclei from each tagmentation reaction are individually distributed into 384-well plates via fluorescence-activated nuclei sorting (FANS) (Supplementary Fig. 3). Each well houses $N$ uniquely first-round indexed nuclei after sorting. Nuclei in the wells are lysed in the pre-loaded buffer containing sodium dodecyl sulphate (SDS) and proteinase K. The lysis process is then quenched, followed by DNA amplification using pre-loaded indexed PCR primers for the second-round barcoding. The PCR products are then pooled for a final round of PCR to add standard Illumina TruSeq adapters, preparing them for next-generation sequencing (NGS) (Supplementary Fig. 2). Using the liquid handler, all steps in 384-well plates can be automated to avoid intricate pipetting.

The accuracy of IT-scATAC-seq was assessed using a species-mixing experiment with mixed human and mouse cell lines. After quality control, the high-quality cells with the number of unique fragments over 2000 were predominantly identified as either mouse ($n = 1784$) or human ($n = 1234$), with only 39 cells identified as doublets, yielding an accuracy rate of 98.72% (Fig. 1b). IT-scATAC-seq was then applied to three human cell lines—HEK293T, H1, and K562—with two replicates per cell line, each containing 384 cells. High correlations in read coverage were observed between replicate libraries for each cell line (Pearson correlation $r > 0.97$) (Fig. 1c). The aggregated single-cell libraries showed strong signal enrichment at TSS and clear nucleosome periodicity patterns (Fig. 1d–f). All input single cells were successfully retrieved. For HEK293T, H1, and K562 cell lines, median unique fragments per cell were 50,276, 23,054, and 23,273, respectively, and the median TSS enrichment scores were 18, 12, and 15, respectively; 100%, 98.7%, and 93.2% out of input cells met the ENCODE's established quality control criteria (TSS score >5 and unique fragments >1000) (Fig. 1e).

Bulk Omni-ATAC-seq was performed in HEK293T cells to evaluate the IT-scATAC-seq profiles further. The bulk libraries exhibited a typical periodic fragment pattern, minimal mitochondrial contamination, high TSS scores, and high FRiP scores (Supplementary Fig. 4), qualifying them as suitable reference libraries. Pseudo-bulk profiles of IT-scATAC-seq libraries (rep #1 and rep #2) showed robust correlations ($r > 0.90$) with bulk libraries(Supplementary Fig. 5a). Additionally, 20 randomly selected single-cell profiles demonstrated high congruence with bulk data with Pearson correlation coefficients ranging from 0.52 to 0.94 (Supplementary Fig. 5a). The aggregated and randomly selected single-cell profiles closely resembled bulk signals in accessible regions and specific loci (Supplementary Fig. 5b). These results confirm the high quality of IT-scATAC-seq libraries regarding the accuracy and signal specificity at the single-cell level.

We merged libraries from cell lines to test IT-scATAC-seq's ability to distinguish different cell types. Using ArchR[22]'s Latent Semantic Indexing (LSI) for dimension reduction, followed by uniform manifold approximation and projection (UMAP) for visualisation, we identified three distinct cell populations corresponding to embryonic stem cells, myeloid cells, and epithelial cells, and the cell identities matched their cell-type encoded barcode (Fig. 1g). While housekeeping locus *GAPDH* loci showed comparable accessibility among all cell types, loci such as *NANOG*, *GATA1*, and *XIST* exhibited strong cell-type specificity (Fig. 1h). Next, the single cells were clustered based on chromVar[23]-calculated bias-corrected deviations (Fig. 1i). This analysis identified cell line-specific transcription factor (TF) motifs: GATA family motifs were enriched in K562 cells, *POU5F1* in H1 cells, and *HOX*, *FOS*, and *JUN* family motifs in HEK293T cells (Fig. 1j, k). Together, these results demonstrate that IT-scATAC-seq is robust in identifying cell types and specific TF motif enrichments.

### Comparing IT-scATAC to other scATAC-seq methods

To further demonstrate the quality of IT-scATAC-seq, we compared its cell line metrics with those of other scATAC-seq methods, including droplet-based scATAC-seq (10X Chromium and Hydrop[24]), microfluidics-based scATAC-seq (Fluidigm C1)[6], plate-based[8], and sci-ATAC-seq[7] and its derivate CH-ATAC-seq[25]. At lower sequencing depths, indicated by a median duplication rate of 54–57% compared to over 95% in plate-based and C1 scATAC-seq (Fig. 2a), IT-scATAC-seq still achieved comparable or higher library complexity, as evidenced by a comparable or higher number of unique fragments per cell (Fig. 2b). While the proportion of sequencing reads mapped to nuclear, but not mitochondrial, DNA was similar (Fig. 2c), IT-scATAC-seq

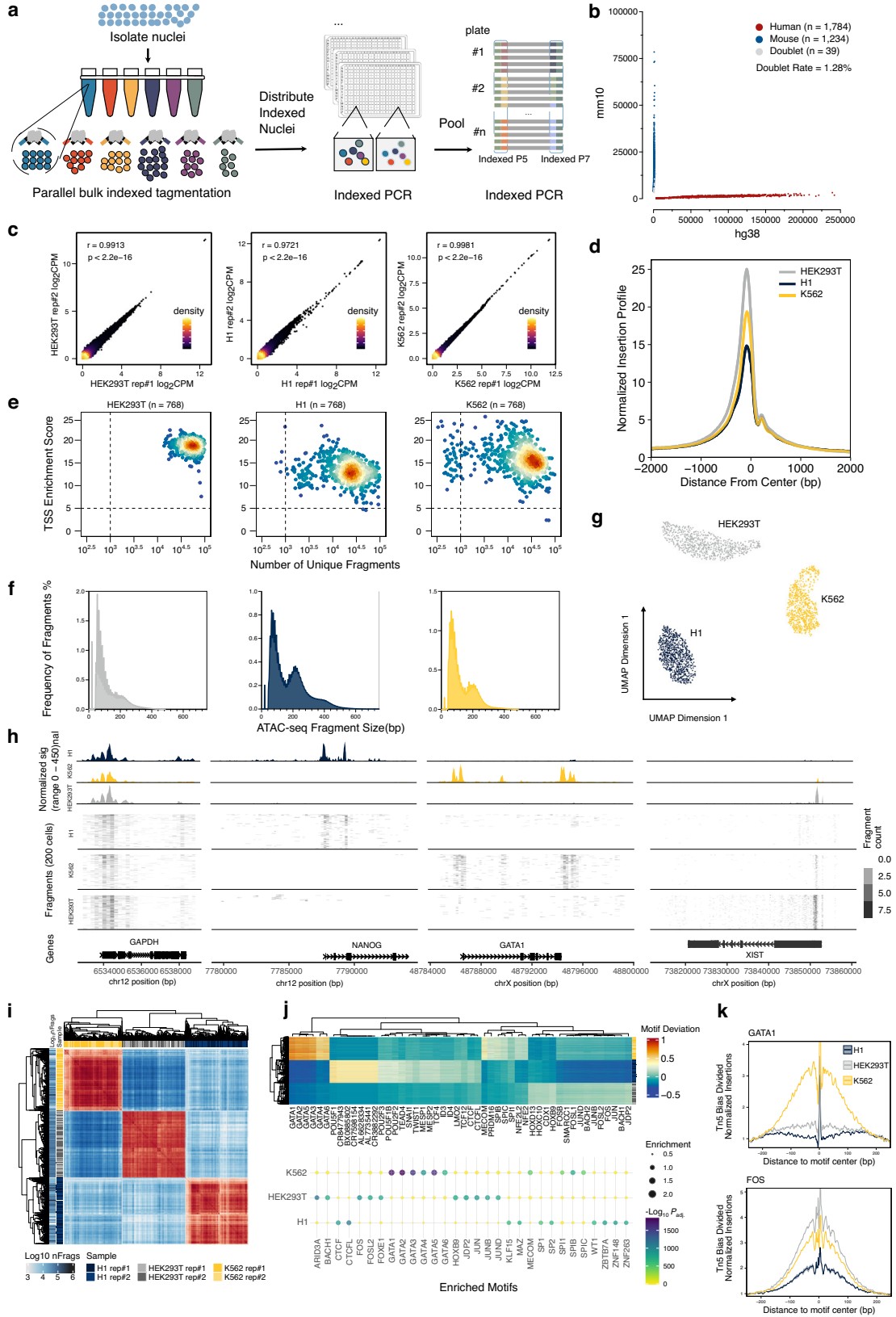

achieved the highest percentage of reads aligned with chromatin accessibility peaks, with a median FRiP score over 65% (Fig. 2d). Additionally, IT-scATAC-seq produced higher or similar median TSS enrichment scores compared with existing methods (Fig. 2e). Unlike the variability seen in the other datasets, IT-scATAC-seq displayed more consistent quality control metrics at the single-cell level, with

data more tightly clustered around the median, indicating stable and consistent single-cell profiles.

Scalability, accuracy, and cost-effectiveness are crucial for implementing scATAC-seq, especially in resource-limited settings. IT-scATAC-seq enhances the throughput by at least one order of magnitude compared to plate-based scATAC-seq[8], increasing cell

**Fig. 1 | Benchmark of IT-scATAC-seq. a** Workflow of IT-scATAC-seq library preparation. Nuclei are isolated and subjected to parallel bulk transposition reactions with indexed Tn5 complexes. The transposed nuclei from each reaction are sorted individually into 384-well plates. After lysis, the first round of barcoded PCR is performed to distinguish cells from different wells. The PCR products are pooled for a second round of barcoded PCR to cover more plates and incorporate the TruSeq adapters. **b** Species mixing experiments for IT-scATAC-seq. Number of unique reads per cell aligning to the human or mouse genome. Cells with less than 90% alignment rate are considered as doublets. **c** Scatter plots showing pairwise Pearson correlation ($r$) in read coverage as $\log_2$ of count per million mapped reads (CPM) across all accessible loci between replicates of IT-scATAC-seq libraries from HEK293T, H1, and K562 cells. Each cell line was profiled with two replicates of 384 single cells, totalling $n = 768$ per cell line. *P-values* were determined using a two-sided Pearson correlation test. **d** Distribution of ATAC-seq signals around ±2 kb from transcription starts sites (TSS) of single-cell aggregates. **e** ATAC-seq insert

fragments frequencies distribution showing nucleosome periodicity of libraries from aggregated single cell profiles. **f** TSS enrichment score plotted against the number of unique fragments for HEK293T, H1, and K562 IT-scATAC-seq libraries. **g** UMAP visualisation of integrated scATAC-seq libraries coloured by cell type identity. **h** Genome tracks displaying aggregated single-cell ATAC-seq signals and per-cell fragment abundance around the *GAPDH, NANOG, GATA1,* and *XIST* loci. **i** Correlation of bias-corrected motif deviations between replicates. **j** Heatmap showing deviations of motifs across single cells (top panel); dot plot displaying motif enrichment, assessed using a two-sided hypergeometric test, with $-\log_{10}$ of *P*-values adjusted for multiple comparisons using the Benjamini–Hochberg method (bottom panel). **k** TF footprinting analysis of GATA and FOS in IT-scATAC-seq signals of three cell lines, normalised for Tn5 insertion bias by dividing the footprint signal by the expected insertion frequency. Source data are provided as a Source Data file.

processing capacity from $10^{2\cdot3}$ to $10^{4\cdot5}$, comparable to the droplet-based scATAC-seq[17] and 10x Genomics scATAC-seq. For accuracy, the doublet rate of IT-scATAC-seq only depends on the accuracy of nuclei sorting (Supplementary Fig. 3), contrasting with droplet-based and sci-ATAC-seq frameworks, which typically have misassignment and barcode collision rates of around 10%[7,17]. IT-scATAC-seq uses parallel bulk tagmentation instead of single-cell individual tagmentation, effectively minimising potential benchtop variations[7,8]. Notably, IT scATAC-seq requires significantly less manual labour than plate-based scATAC-seq[8]. For example, capturing 5000 cells with plate-based methods requires handling at least ten 384-well plates, whereas IT-scATAC-seq achieves this with just a single plate. Furthermore, using the liquid handling system significantly reduces complex and labour-intensive pipetting and lowers the risk of primer cross-contamination during PCR. Library preparation for 10,000 cells can be completed within a single day (Supplementary Fig. 6a). Although sorting is time-consuming, most processes are automated (Supplementary Fig. 6b). As for reagent cost, IT-scATAC-seq significantly reduces the per-cell cost by up to 100 times, depending on the number of cells profiled (the more cells processed, the lower the cost per cell). As a result, the library preparation cost is substantially reduced to ~$0.01 per cell (Supplementary Fig. 6c), making it considerably more cost-effective than many scATAC-seq methods (Supplementary Table 1)[16]. Moreover, all reagents required for IT-scATAC-seq are listed and can be readily prepared in-house (Methods). Overall, IT-scATAC balances single-cell omics' sensitivity, accuracy, precision, throughput, and cost-effectiveness, providing a strategy for high-quality single-cell chromatin accessibility profiling (Supplementary Data 2).

## IT-scATAC-seq detects high plasticity of cell fate during early embryogenesis

Naïve mouse embryonic stem cells (ESCs) were subjected to a two-day differentiation period to primed epiblast-like stem cells (EpiLCs), a transient interval that has already acquired competence for differentiating towards downstream mesodermal (Meso), endodermal (Endo) and ectodermal (Ecto) lineages (Fig. 3a). Analysis of the EpiLCs IT-scATAC-seq library showed that 4167 passed the quality control (Fig. 3b). From these cells, we harvested a total of 131.81 million fragments, with the fragment size distribution displaying a typical nucleosomal pattern and an enrichment of signal around the TSS region (Fig. 3c, d). With a sequencing depth marked by a 44% duplication rate (Fig. 3e), IT-scATAC-seq demonstrated a 98% read alignment rate and a median of 18,058 unique fragments per cell, confirming high library complexity (Fig. 3f, g). Additionally, cells demonstrated an average TSS enrichment score of 14.35, low mitochondrial contamination (median 1.62%) and a high FRiP score (median 0.69) (Fig. 3h–j). These results showed the high quality of the IT-scATAC-seq library.

Previous research demonstrated that the EpiLCs are competent to differentiate into all three germ layers[26]. However, the mechanisms by which naïve ESCs transit to EpiLCs and how gene cascades are selectively activated to determine the cell fate have not been fully elucidated by scRNA-seq alone[27]. We used gene activity scores[22], which quantify chromatin accessibility around genes weighted by distance and size, normalised across the gene region, to infer potential regulatory impacts on gene expression. We leveraged a targeted panel of lineage marker genes[27,28] and calculated lineage scores for each cell based on the average marker activity following the same strategy previously described in itChIP-seq(see Methods)[29]. Unsupervised clustering of 4167 single cells based on normalised accessibility profiles of lineage-specific markers identified 10 distinct clusters (Fig. 3k). These clusters entailed a spectrum of cellular state, ranging from naïve ESCs with pronounced accessibility in ESC marker regions to cells exhibiting increased accessibility across both ESC and multiple lineage markers, suggestive of priming for germ layer differentiation, and cells with commitment to specific germ layers (Fig. 3k).

Notably, a substantial number of cells occupied intermediate states, including those with relatively low accessibility for all four categorical markers compared with primed ESCs, indicative of formative ESCs (Fig. 3k). Additionally, cells with transitional combinations of marker gene accessibility, such as meso-endo-ecto ($n = 171$) and endo-ecto ($n = 120$), underlined the multifaceted nature of naïve to epiblast-like transition. Pseudo-temporal trajectory plots, comparing germ-layer scores against ESC scores, revealed an increase in mesoderm accessibility as cells transitioned from the naïve ESC state; in contrast, endoderm and ectoderm accessibility scores remained relatively stable or slightly declined before the loss of pluripotency (Fig. 3l), suggesting a potential epigenetic restriction prior to definitive lineage commitment. These observations collectively echoed the concept of cell fate plasticity, where cells exhibit the potential to transit between states and adapt to developmental cues through dynamic epigenetic remodelling[30–32]. The simultaneously increased accessibility of multiple lineage markers post-ESC state exiting substantiated this plasticity, highlighting the cells' adaptability and the non-fixed nature of development. These findings underscore the importance and effectiveness of advanced high-resolution single-cell technologies, such as IT-scATAC-seq, for dissecting the regulatory mechanisms that govern dynamic and transient cell states.

## IT-scATAC-seq distinguishes the cellular heterogeneity across human PBMCs

To evaluate IT-scATAC-seq's ability to resolve diverse cell types and dissect epigenomic heterogeneity, we applied it to cryopreserved PBMCs collected from healthy donors during routine hospital check-ups. These samples provide a more physiologically relevant setting compared to cell lines. We additionally incorporated two published healthy PBMC scATAC-seq datasets[16], generated using different

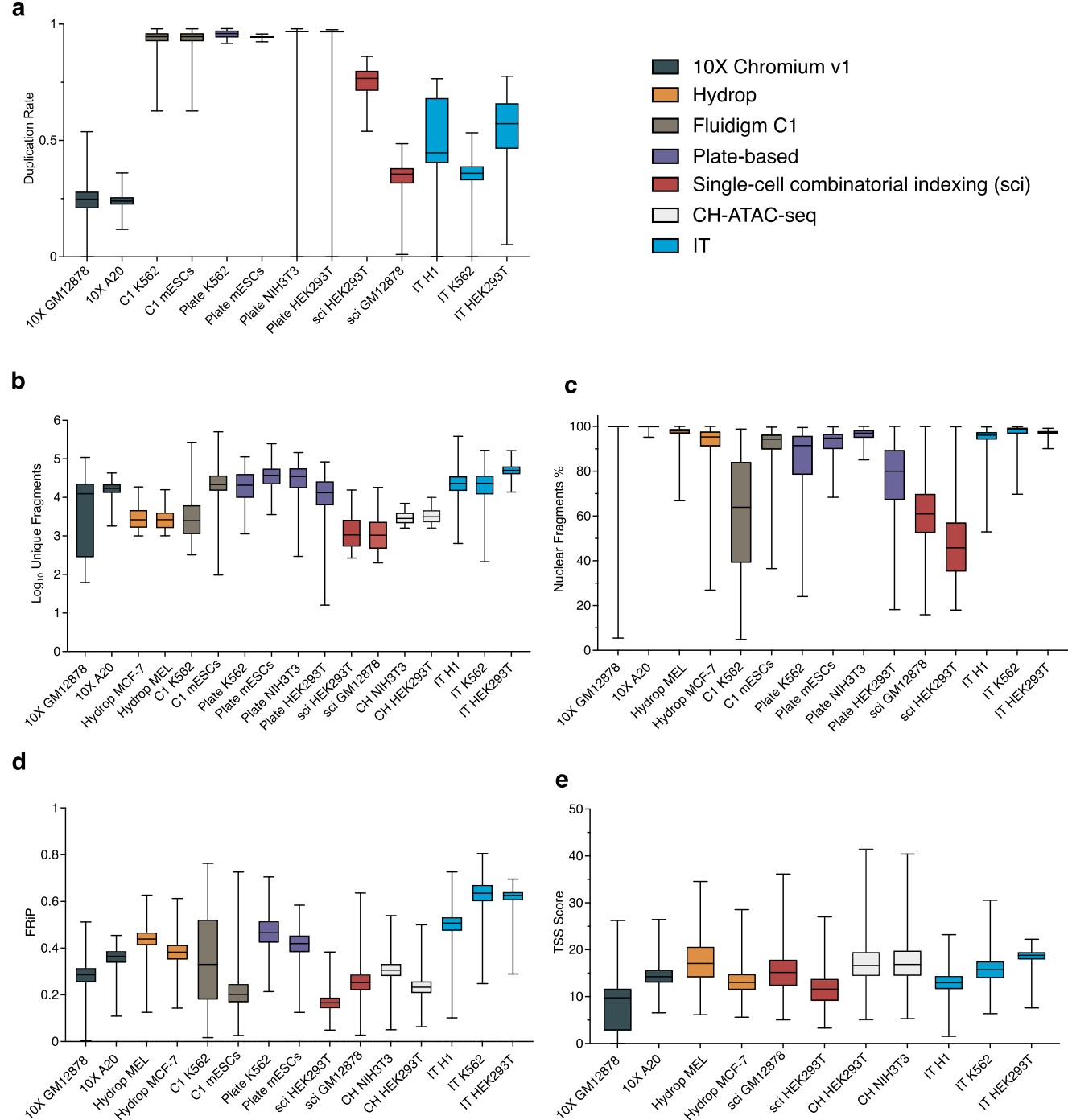

**Fig. 2 | Comparison of IT-scATAC-seq with other scATAC-seq methods in cell lines.** Box plots show duplication rate (**a**), an indicator of sequencing depth, library complexity (**b**), measured as log$_{10}$ of unique fragments per cell, percentage of fragments mapped to the genome (**c**), FRiP per cell (**d**), and TSS enrichment score (**e**) across different methods, where the centre line represents the median, the box bounds indicate the interquartile range (IQR, 25th to 75th percentile), and the whiskers extend to the minimum and maximum values. The number of single

cells analysed for each method and cell line are as follows: 10X GM12878 ($n$ = 996), 10X A20 ($n$ = 474), Hydro MCF-7 ($n$ = 889), Hydro MEL ($n$ = 461), CH NIH3T3 ($n$ = 2083), CH HEK293T ($n$ = 2846), sci HEK293T ($n$ = 343), sci GM12878 ($n$ = 1197), Plate K562 ($n$ = 192), Plate mESCs ($n$ = 192), Plate NIH3T3 ($n$ = 139), Plate HEK293T ($n$ = 172), C1 K562 ($n$ = 192), C1 mESCs ($n$ = 192), IT H1 ($n$ = 767), IT K562 ($n$ = 766), IT HEK293T ($n$ = 768). Source data are provided as a Source Data file.

technologies – 10X Genomics v2.1 (10X) and s3-ATAC (s3) – for comparative analyses (Supplementary Fig. 7a).

From our IT-scATAC-seq library, a total of 7628 single cells passed quality control, exhibiting clear nucleosome banding patterns and strong signal enrichment around the TSS region(Supplementary Fig. 7a–e). Although the sequencing depth was relatively

modest (-10,000 reads per cell)—yielding a lower median number of unique fragments (3026.5) than those reported for 10X (13,771) and s3 (15,395)[33]—the IT-scATAC-seq PBMC profiles showed higher signal specificity, as shown by higher median TSS enrichment (25.03 vs. 16.16 and 7.39, respectively) and higher median FRiP (0.54 vs. 0.52 and 0.23, respectively) (Supplementary Fig. 7d–f). After LSI

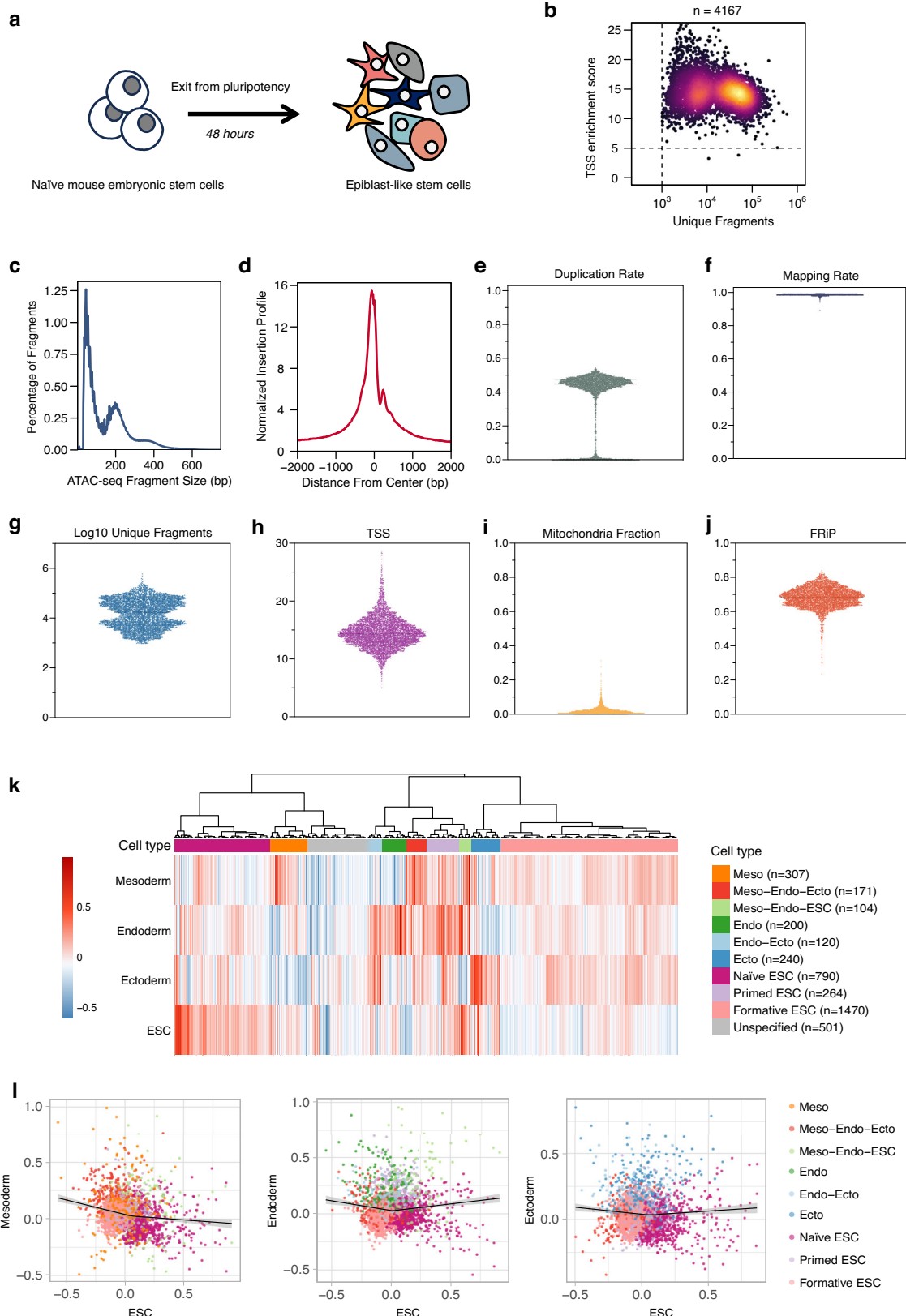

dimension reduction and batch effect correction[34], cells from all three datasets were dispersed throughout the UMAP space, revealing 14 distinct immune cell populations (Fig. 4a and Supplementary Fig. 7g). This suggests that IT-scATAC-seq robustly captures a wide spectrum of immune cell lineages in a manner comparable to established scATAC-seq platforms.

To refine and merge clusters, We integrated single-cell gene expression hemopoiesis scRNA-seq datasets[35,36], identifying nine major cell clusters from all PBMCs: B cells ($n = 1601$), basophils ($n = 257$), CD14 monocytes ($n = 2044$), CD16 monocytes ($n = 631$), CD4 memory T cells ($n = 5035$), CD4 naive T cells ($n = 2257$), CD8 memory T cells ($n = 1701$), CD8 naive T cells ($n = 2279$), natural killer cells ($n = 3739$),

**Fig. 3 | IT-scATAC-seq revealed a high degree of cell fate plasticity during mouse early embryogenesis. a** Schematic images showing naïve mouse embryonic stem cells (ESCs) were undergone a 48-h differentiation and were subjected to IT-scATAC-seq profiling. **b** $Log_{10}$ of number of unique fragments plotted against the TSS score, and cells within the upper right quadrant ($n = 4167$) passed QC and were subjected to downstream analysis. **c** Fragment size distribution. **d** Enrichment of ATAC-seq signals up and downstream 2 kb to the TSS region (**d**). Violin plots of per cell ($n = 4167$) duplication rate (**e**), read alignment rate (**f**), $Log_{10}$ of number of unique fragments (**g**), TSS enrichment score (**h**), mitochondrial fraction of total uniquely mapped reads (**i**), and fraction of reads in peaks (FRiP) (**j**). **k** Heatmap displaying 10 groups of 4167 single cells clustered based on lineage scores for ESC, mesoderm (Meso), endoderm (Endo), and ectoderm (Ecto), where lineage scores for each cell were calculated using gene activity scores of marker genes specific to each lineage (see Methods). **l** Scatter plots of ESC lineage scores plot against Meso, Endo, and Ecto lineage scores for each single cell. The regression line was fitted using generalized additive model, and the shaded band represents the 95% confidence interval of the fitted line. Source data are provided as a Source Data file.

conventional dendritic cells (cDC, $n = 188$), and progenitors ($n = 161$) (Fig. 4b). The per-cell gene activity overlay on the UMAP embedding showed consistent aggregated accessibility for cell-type-specific genes such as *PAX5, MS4A1* and *EBF1* for B cells; *CD3G, IL7R, CD8A* for T cell lineages; *CD16a (FGCR3A), NKG7* and *IL2RB* for NK cells; and *CD14, CEBPB*, and *CCR2* – corresponded well with the identified cluster identities (Fig. 4c).

To explore the regulatory landscape underlying these cell types, we called peaks using pseudo-bulk replicates from the nine cell types, creating a union set of 185,353 reproducible accessible peaks and identifying 64,606 differential peaks across cell types (FDR ≤ 0.1 & $log_2$ fold change >1) (Fig. 4d and Supplementary Data 3). We next examined the enrichment of TF-binding motifs (FDR ≤ 0.1 & $log_2$ fold change >0.5) within the differentially accessible regions across major PBMC populations. We showed their distinct transcriptional programs (Fig. 4e and Supplementary Fig. 8). Members of the IRF and ETS families (including *SPI1* and *SPIB*), along with *BCL11*, displayed strong enrichment in myeloid lineages, cDCs, and B cells, with IRF prominent in B cells. By contrast, the C/EBP family (*CEBPA, CEBPB, CEBPD, CEBPE, CEBPG*) exhibited significant enrichment only to monocytes. In NK and CD8 Memory T cell subsets, we observed characteristic T-box (*TBX4, TBX5, TBX10, TBX20*) and RUNX (*RUNX1, RUNX2*) motifs, reflecting their regulatory impact on cytotoxic functions. Meanwhile, naïve T cells showed TCF-family motif variability (e.g., *TCF7, TCF7L1, LEF1*), highlighting TF networks that maintain an undifferentiated state and govern T cell receptor repertoire. Basophils were marked with GATA-family motifs, which aligns with GATA-driven regulation of their generation and activation. Notably, the Sp-family, C/EBP family, *BCL11, FOS, JDP2, NFE2* and *NF-Y* were identified as key TFs driving lineage-specific differences (Fig. 4f and Supplementary Data 3). These findings were in concordance with those observed at the single-cell transcriptome[16,37,38] and bulk scale[39], indicating that IT-scATAC-seq can effectively distinguish and characterise cell type-specific gene regulatory programs. These results further validate IT-scATAC-seq as a scalable and cost-effective platform for single-cell chromatin accessibility profiling when applied to clinical samples, capable of resolving immune cell heterogeneity with high fidelity.

## Discussion

Single-cell chromatin accessibility profiling has become a critical tool for understanding gene regulation, cellular heterogeneity, and epigenomic dynamics. Here, we introduce IT-scATAC-seq, a cost-effective, scalable and robust method that enables high-throughput single-cell chromatin accessibility profiling at lower per-cell cost. The IT-scATAC-seq process involves four main steps: (1) assembly of indexed Tn5 transposome complex, (2) parallel bulk nuclei tagmentation, (3) sorting different indexed nuclei into the same well for barcoded PCR, and (4) pooling and PCR for Illumina Truseq adapter addition. This streamlined workflow allows for $10^4$ cells to be completed within one day.

Through benchmarking analyses, IT-scATAC-seq demonstrated high library complexity, strong enrichment at TSS, and low mitochondrial contamination. The overall data quality is either comparable to or exceeds established plate-based scATAC-seq[8] and

commercial 10x Genomics ATAC-seq[16]. To validate the method's broad applicability, we applied IT-scATAC-seq to mouse embryonic stem cell (mESC) differentiation and human peripheral blood mononuclear cells (PBMCs). During mESC differentiation, chromatin accessibility profiles revealed an intermediate state where cells exhibited accessibility at both pluripotency and lineage-specific regulatory elements, suggesting a dynamic priming process during cell-fate commitment. These findings align with the concept of cell-fate plasticity, highlighting the gradual and coordinated chromatin remodelling that occurs during early embryogenesis. In cryopreserved human PBMCs, IT-scATAC-seq successfully resolved immune cell subsets, demonstrating its ability to capture epigenomic heterogeneity in complex primary tissues. These results confirm that IT-scATAC-seq is well-suited for profiling chromatin accessibility across diverse biological systems.

Compared with existing scATAC-seq methodologies, IT-scATAC-seq balances cost efficiency, scalability, and data quality (Supplementary Table 1). By implementing parallel bulk tagmentation with indexed Tn5 transposases, IT-scATAC-seq significantly reduces per-cell reagent consumption, achieving a cost of ~$0.01 per cell, which is lower than the single-cell tagmentation-based method. Unlike sci-based methods, it does not require assembling many indexed Tn5 complexes, simplifying the workflow while maintaining high library complexity. Unlike commercial platforms such as 10X Genomics, Fluidigm C1[6], and Takara ICELL8[9], IT-scATAC-seq does not require specialised single-cell instrumentation, making it compatible with standard laboratory equipment. Compared to the plate-based approach, which has been demonstrated to be robust and accessible to most laboratories[8,40], IT-scATAC-seq significantly enhances throughput and processing efficiency. Analysing thousands to tens of thousands of cells is now achievable at reduced labour and consumable costs. Optionally, the automated liquid handling system can be used during the second indexing step to reduce intricate pipetting, thereby substantially mitigating the risk of primer cross-contamination. Furthermore, TruSeq-compatible library design ensures broad sequencing compatibility and significantly lowers sequencing costs, making IT-scATAC-seq a practical solution for large-scale epigenomic studies.

While IT-scATAC-seq offers several advantages, it also has some trade-offs, primarily due to its in-house nature. First, indexed Tn5 transposase may be a barrier for labs without enzyme preparation capabilities, though commercial Tn5 is available. Second, although IT-scATAC-seq simplifies the workflow, its cell throughput—given equivalent time and labour— is lower than sci-ATAC-seq[7] and its derivatives like EasySciATAC[20]. Third, IT-scATAC-seq improves resolution and lowers barcode misassignment through FANS but requires flow cytometry resources and constitutes the most time-consuming stage of the workflow. Future optimisations could develop alternative nuclei-handling strategies to reduce FANS dependency and improve throughput.

Beyond its current applications, IT-scATAC-seq holds the potential for expanding its compatibility with other single-cell multi-omics platforms and multimodal integration. For example, the IT workflow could be integrated with single-cell whole genome sequencing[41],

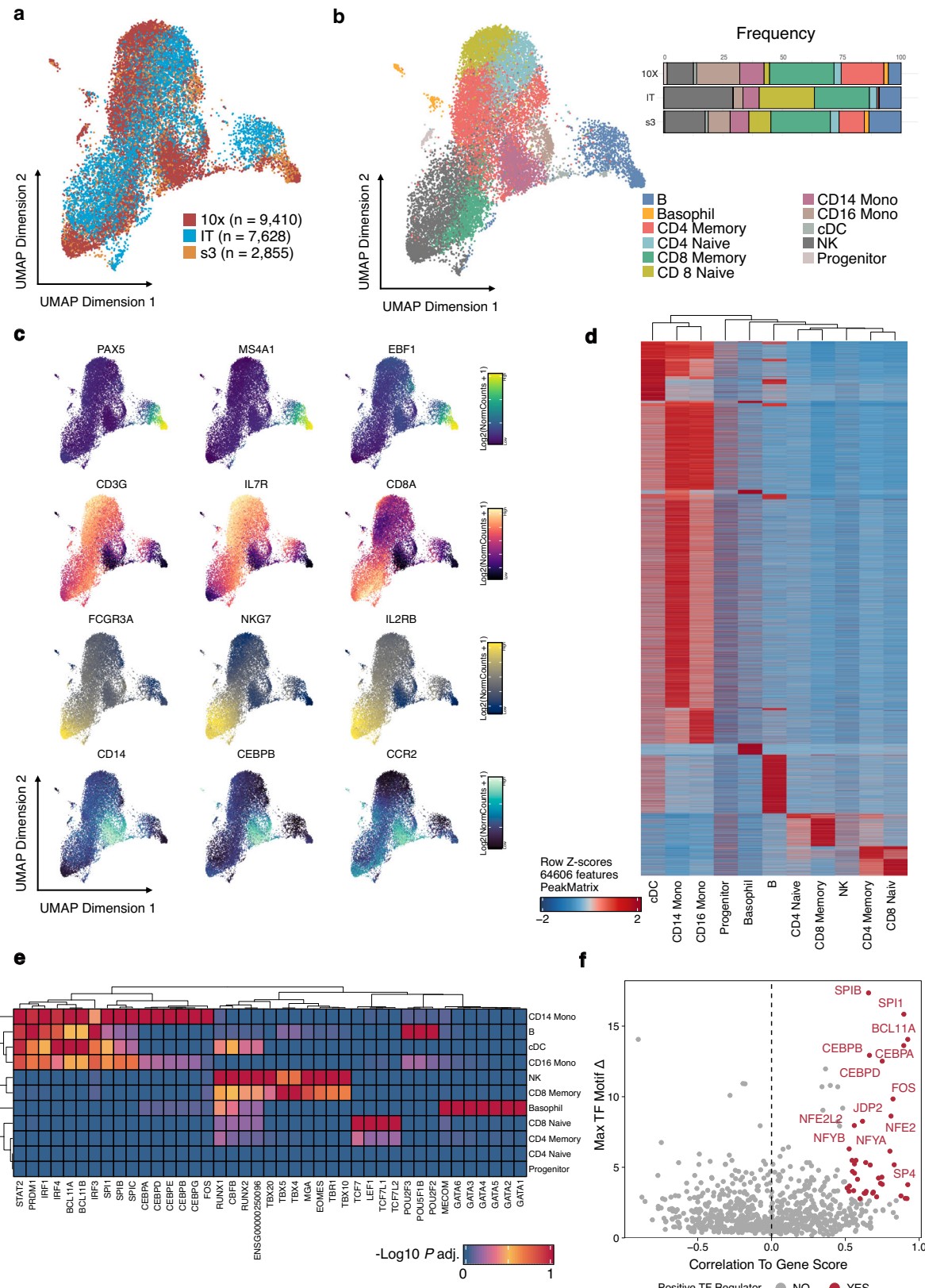

CUT&Tag[42] and HiC-seq[43], to extend its capabilities in single-cell epigenomics. Additionally, IT-scATAC-seq could be adapted for simultaneous chromatin accessibility and transcriptomic profiling (e.g., scATAC-seq + scRNA-seq), enabling a more comprehensive view of gene regulatory networks at the single-cell level[10,13–15,44–46], offering comprehensive insights into cellular function and regulation.

In summary, IT-scATAC-seq represents a robust, cost-effective, and scalable alternative to existing scATAC-seq methods. It provides high-quality single-cell chromatin accessibility data while eliminating the need for specialised microfluidic instruments. While limitations such as indexed Tn5 production, lower throughput compared to EasySciATAC, and reliance on FANS should be

**Fig. 4 | IT-scATAC-seq dissects cellular heterogeneity in human PBMCs. a** UMAP plots showing IT-scATAC-seq profiles of PBMC samples from healthy donors (IT, $n = 7628$), and two additional PBMC scATAC-seq datasets from healthy donors profiled by 10X (10X, $n = 9411$) and s3 (s3, $n = 2855$), coloured by sample origin. **b** UMAP visualisation coloured by cell type identity, including B cells, NK cells, T cells, monocytes and progenitors with a top panel showing cell type fractions for each sample. **c** Lineage-specific markers overlaid on the UMAP embedding, including *PAX5*, *MS4A1*, and *EBF1* for B cells; *CD3G*, *IL7R*, and *CD8A* for T cells; *FCGR3A* (*CD16*), *NKG7*, and *IL2RB* for NK cells; and *CD14*, *CEBPB*, and *CCR2* for monocytes. Visualisation is coloured by normalised gene scores, with the MAGIC algorithm used to smooth drop-out noise. **d** Heatmap showing Z-scores of normalised chromatin accessibility for 64,606 cell identity-specific marker peaks (FDR ≤ 0.1, log₂ fold change >1) identified across scATAC-seq clusters using the two-sided Wilcoxon rank-sum test, with *P*-values

adjusted for multiple comparisons using the Benjamini–Hochberg correction. **e** Heatmap displaying the top 10 TF binding motifs with the highest variability in respective marker peaks of each cluster ranked by adjusted *P*-value, calculated using a two-sided hypergeometric test, and *P*-values adjusted for multiple comparisons using the Benjamini–Hochberg method. **f** Scattered plot illustrating the correlation between motif accessibility and gene expression. Each point represents a TF, with the x-axis showing the correlation to gene expression and the y-axis indicating the maximum TF motif delta (variability) across clusters. *P*-values were derived from a two-sided Pearson correlation test and adjusted for multiple comparisons using the Bonferroni method. TFs identified as positive regulators (correlation >0.5 and adjusted *P*-value < 0.01, with max delta in the top quartile) are highlighted in red and other TFs in grey. Source data are provided as a Source Data file.

considered, its strengths in data resolution, cost efficiency, and accessibility make it a valuable tool for single-cell epigenomics research. Further optimisations could enhance its automation and scalability, expanding its applications to developmental biology, and clinical genomics.

## Methods

### Cell culture

The human HEK293T and mouse NIH/3T3 were ordered from ATCC and routinely maintained in High-glucose Dulbecco's modified Eagle's medium (DMEM) containing 10% Fetal Bovine Serum (FBS) and 1% Penicillin/Streptomycin. The K562 cells were purchased from ATCC and maintained in an IMDM medium containing 10% FBS. The H1 was obtained from WiCell Research Institute (WA01). Cells were cultured in Essential 8 with ROCK inhibitor Y-27632 (HY-10071, MedChemExpress) on plates pre-coated with Matrigel (Corning). The medium was refreshed daily, and the cells were passaged with Accutase (Gibco) every 3 days. The B6 murine ESCs were obtained as a gift from Pengtao Liu's lab at the University of Hong Kong (HKU) and cultured on gelatin-coated dishes in 2i medium composed of High-glucose DMEM supplemented with 15% stem-cell qualified FBS, 2 mM GlutaMAX, Non-essential amino acids (NEAA), 0.1 mM β-mercaptoethanol, 1000 U/ml recombinant mouse LIF (ESG1107, Merck Millipore), 2i 1 μM PD032591 and 3 μM CHIR99021(HY-10254 and HY-10182, MedChemExpress) and 1% Penicillin/Streptomycin. The basic medium and supplements for cell culture were purchased from Thermofisher. All the cells were cultured at 37 °C in 5% CO2 and tested negative for mycoplasma infection using the PCR method by the Centre for PanorOmic Sciences, HKU.

### Purification of transposase Tn5

The pTBX1-Tn5 plasmid was purchased from Addgene (60240). Briefly, pTBX1-Tn5 plasmid was transformed into competent *E. coli* C3013 cells (C2527I, NEB) and induced with 250 μL 1 M Isopropyl β- d-1-thiogalactopyranoside (IPTG) at 23 °C for 5 hours. Cell pellet was resuspended in 60 ml HEGX buffer (20 mM HEPES buffer pH 7.2, 1.0 M NaCl, 1 mM ethylenediaminetetraacetic acid (EDTA), 10% v/v glycerol, 0.2% v/v triton X-100 and 10 mM PMSF) and sonicated using Covaris sonicator with 10 cycles of 30 s on and 30 s off, 40% duty. The cleared Tn5-CBD protein fraction was enriched with chitin resin (S6651S, NEB) in the cold room for 2 hours and further washed with 200 ml of HEGX buffer. The Tn5 protein was released by 100 mM dithiothreitol (DTT) cleavage, concentrated with Pierce™ Protein 30 K MWCO Concentrators and dialysed twice in 1 L 2X HEPES dialysis buffer (100 mM HEPES pH 7.2, 0.2 M NaCl, 0.2 mM EDTA, 20% w/v glycerol, and 2 mM dithiothreitol (DTT). After dialysis, the Tn5 was equilibrated with pure glycerol to 60% concentration. The final Tn5 was quantified by SDS-PAGE and Coomassie Blue Staining using the fitting curve plotted by standard BSA. Tn5 was quantified as 1.6 μg/μL, approximately 30 μM in this study.

### Preparation of indexed Tn5 transposome complex

Dissolve the indexed adapters and Tn5 reverse adapters (ordered from IDT; sequences provided in Supplementary Data 1) with annealing buffer (10 mM Tris-HCl pH 8.0, 50 NaCl, 2 mM EDTA) to make 200 μM stock. Prepare 15 μL of individual adapter with 15 μL reverse adapter in 200 μL PCR tube and anneal in a thermocycler as follows: 98 °C for 10 min, and slowly cool down to 23 °C with −0.1 °C/s. Mix the annealed adapter with 100 μL 30 μM Tn5 and 70 μL coupling buffer (100 mM HEPES-NaOH, 500 mM NaCl, 50% v/v Glycerol, 0.5 mM EDTA, 2 mM DTT), and incubate in thermomixer at 25 °C, 1000 rpm for one hour. The indexed Tn5 transposome was prepared by mixing 20 μL of the paired two Tn5-adapters with 80 μL coupling buffer, and the resulting Tn5 transposome complex was 5 μM and can be stored at −20 °C without activity loss for more than one year.

### Quality control of assembled transposases

Prepare 1 μL 300 ng/μL genomic DNA, 4 μL 5xTAPS-DMF buffer (50 mM TAPS-NaOH pH 8.2, 25 mM MgCl2, 50% DMF), 13 μL H2O, 2 μL assembled Tn5. Incubate at 55 °C for 10 min, then add 2 μL 10X STOP buffer (2% SDS, 40 mM EDTA) and quench at 37 °C 15 min to dissociate Tn5 from DNA. Add 5 μL 6x Loading dye and run 1.5% DNA gel. The majority of tagmented DNA sizes were less than 1000 bp, indicating the assembled transposases are qualified for downstream experiments. In this study, we randomly picked indexed Tn5 for quality assessment.

### mESCs-epiblast differentiation

mESCs were cultured in 2i medium to 60–80% confluency and dissociated into single cells using 0.1% Trypsin. After washing twice with PBS buffer, the mESCs were then resuspended in fresh embryoid body media (withdraw of 2i and mLIF) and seeded on a gelatin-coated plate. The spontaneously differentiated cells were collected on day 2 and ready for IT-scATAC-seq.

### Isolation of human peripheral blood mononuclear cells (PBMCs)

The study design and conduct complied with all relevant regulations regarding the use of human study participants, approved by Dongguan Children Hospital and The University of Hong Kong, and was conducted in accordance with the criteria set by the Declaration of Helsinki. About 5 mL of blood was taken from two healthy donors, with informed consent and human tissue procurement under the guidance of ethical regulations of Dongguan Children Hospital and the University of Hong Kong. The PBMCs were isolated using Ficoll-Paque-based gradient separation and frozen in liquid nitrogen until usage. The frozen cells were rapidly thawed in a water bath at 37 °C and transferred to a 15 ml tube containing 10 mL prewarmed medium. The cell suspension was centrifuged at $500 \times g$ for 5 min at room temperature to pellet down. The cell pellet was resuspended in 1 mL prewarmed medium, and 10 μL was taken to check the cell viability. PBMCs from two healthy donors were mixed. Considering that some cells in PBMCs were fragile and easily break up upon nuclei isolation, we

gently fixed the cells with 0.2% formaldehyde at room temperature for 5 minutes, which was then quenched with 125 mM glycine before nuclei isolation for IT-scATAC-seq.

## IT-scATAC-seq library preparation

Nuclei were prepared following Omni-ATAC protocol and resuspended in 0.33x PBS buffer. Next, 76 μL nuclei (~5 × 10$^4$) were aliquoted to several 1.5 ml Eppendorf DNA LoBind ® Tubes, and 20 μL 5xTAPS-DMF buffer and 4 μL 5 μM indexed Tn5 transposome complex were added. The tagmentation reactions were performed on a thermomixer at 37 °C, 500 rpm for 30 min. Then, 500 μL stop buffer (1xPBS, 1% BSA and 20 mM EDTA) was added to quench the reaction on ice for 10 min and transferred to FANS tubes. DAPI was added at a final concentration of 1 μg/mL to stain the nuclei before sorting. During the tagmentation, 350 nL lysis buffer (10 mM Tris-HCl, 10 mM NaCl, 0.2% SDS and 0.2 μ/ml Proteinase K) were distributed to 384-plates by Echo® 550 Liquid Handler (Labcyte) and centrifuge at 3000 × g for 3 min. Different index-tagmented nuclei can be sorted into the same well. After sorting, the plates were centrifuged at 3000 × g for 3 min and the nuclei were lysed at 55 °C for 10 min and 100 nL 10% Triton X100 was added to quench SDS. Then, 25 nL 20 μM indexed forward and reverse primers (H5XX and H7XX), as well as 0.5 μL High-Fidelity 2X PCR Master Mix (M0494L, NEB), were added to each well. The first round of amplification was performed following 72 °C 5 min, 98 °C 30 s; 12 cycles of 98 °C 20 s, 63 °C 30 s; 72 °C 1 min; 72 °C 5 min, 4 °C hold. The PCR product was pooled by centrifuge, followed by purification using MinElute PCR purification kit and eluted with 50 μL nuclease-free H2O. The undesired fragments, primers and adapters were removed by Exo I digestion (M0293S, NEB), 1.0 x AMPure XP beads selection, and eluted with 25 μL nuclease-free H$_2$O. The Truseq P5/P7 adapters containing different barcoded primers were added by the second PCR with another 2–3 amplification cycles. After another double AMPure XP beads selection (0.5×/0.35×), the libraries were sent for quality control and NGS by ANOROAD GENOME. A step-by-step protocol is also deposited at protocols.io named IT-scATAC-seq (DOI: dx.doi.org/10.17504/protocols.io.5jyl8d4wrg2w/v1).

## Bulk Omni-ATAC-seq processing and visualisation

Quality control of bulk HEK293T ATAC-seq data was processed following Omni-ATAC protocol[47]. Briefly, cutadapt[48] 4.5 was used to remove Nextera adapters at both 5'- and 3'-end of each read. The trimmed reads were mapped to the human GRCh38 genome using BWA-MEM[49] v.0.7.17. *MarkDuplicates* of Picard Tools 3.1.0 was used to mark and remove duplicated reads. *CollectInsertSizeMetrics* of Picard Tools 3.1.0 were used to calculate the fragment size. Deeptools[50] (version 3.5.2) were used to compute the matrix and plot the heatmap to visualise the enriched signal around ±5 kb up and downstream to the TSS region and to estimate the TSS score.

## Single-cell ATAC-seq data pre-processing

Cutadapt[48] 4.5 was used to remove TruSeq Index 1 (i7) Adapters and Index 2 (i5) Adapters at both 5'- and 3'-end of each read. The barcode sequences were then extracted from 5'-end of each read sequence and appended to read headers of the paired-end reads by Cutadapt 4.5 with *--rename=CB:Z: (r1.adapter_name)(r2.adapter_name) -e 0.01 --no-indels --action=trim*, and adapter sequences and name are specified in FASTA files with parameters -g and -G. The trimmed and barcode-extracted reads were mapped to the corresponding reference genomes, including human (GRch38) for HEK293T and human PBMCs, and human (GRCh38) and mouse (mm10) hybrid genome assembly for species-mixing experiments, using BWA-MEM[49] v.0.7.17. The bam file is then sorted by the cell barcode (CB) tag and split into BAM file by CB using SAMtools[51] 1.17. *MarkDuplicates* of Picard Tools 3.1.0 was used to mark and remove duplicated reads for the demultiplexed BAM file for each single cell. The deduplicated BAM were then merged using

SAMtools into a deduplicated single-cell aggregate BAM file for downstream analysis. Using deduplicated single cell aggregates BAM file, accessible chromatin regions (peaks) were called using MACS2[52], with parameters *-f BAMPE -g hs --shift -75 --extsize 150 --nomodel --call-summits --nolambda --keep-dup all -p 0.01 -B*.

Bamcoverage of Deeptools suite (version 3.5.2) was used first to normalise total reads to 10,000,00 and generate BigWig and Bedgraph files with the parameters *--scaleFactor 10,000,000/reads_number --binSize 50*. We used Deeptool's *multiBigwigSummary* and *plotCorrelation* to calculate the Pearson correlation coefficient between the normalised single-cell aggregate, randomly selected single-cell profiles and bulk Omni-ATAC-seq of HEK293T.

## Species mixing experiments data analysis

For the BAM file generated for each single cell, SAMtools idxstats were used to calculate the fraction of reads mapped to human (GRCh38) and mouse (mm10) genomes. Cells with over 2000 unique fragments were retained as high-quality cells. If the fraction mapped to the human genome >0.90, the cell was identified as a human cell; if the fraction mapped to the human genome 0.10, the cell was identified as a mouse cell; the cell otherwise is classified as a doublet.

## IT-scATAC-seq library quality control

Deeptools *bamCoverage* and *multiBigwigSummary* was used to calculate the normalised coverage of single-cell aggregates of each sample in CPM with default parameter. Deeptools *outRawCounts* was used to generate raw metrics for calculating the Pearson correlation coefficient (*r*) for the replicates of the single-cell libraries. *CollectInsertSizeMetrics* of Picard Tools 3.1.0 were used to calculate the fragment size of single-cell aggregates' libraries. The duplication rate was estimated using the metric file generated by Picard *MarkDuplicates*. SAMtools *idxstats* was used to calculate the number of unique fragments and the percentage of mitochondrial fragments. Sinto (0.10.0, https://timoast.github.io/sinto) were used to generate fragment files from the single cell aggregates BAM file. The fragment file was imported to ArchR[22] to generate Arrow files and obtain quality control data, including the number of unique fragments per cell, TSS enrichment score, and FRiP. Aggregated single-cell ATAC-seq signal and per-cell fragment abundance were plotted using Signac[53].

## Comparison with existing scATAC-seq methods performed on cell lines

Quality control metrics were obtained from plate-based methods to compare quality control metrics with the plate-based and C1-based methods[8](https://github.com/dbrg77/plate_scATAC-seq). 10X GM12878 and A20 Cells quality control metrics were obtained from www.10xgenomics.com/datasets/. For other scATAC-seq methods, fragments files of the following dataset were downloaded from the Gene Expression Omnibus (GEO) or website, including sci[54] (GSM2970932), CH-ATAC-seq[25](https://bis.zju.edu.cn/chatac/), HydropATAC[24] (GSM5343842), and imported to ArchR to calculate the quality control metrics.

## Analysis of IT-scATAC-seq EpiLC library

The BAM file of deduplicated single-cell aggregates was converted to a fragment BED file with Tn5 insertion centering correction using fragment function in Sinto 0.10.0 with parameters *--collapse_within*. The fragment file was compressed using bgzip 1.18 and indexed by tabix 1.18. ArchR[22] was used to create an Arrow file using the fragment file; the quality control criteria were set as TSS > 5 and a number of unique fragments >1000. The TSS enrichment score for each single cell was calculated at the same time. When creating the Arrow file, a Title Matrix counting the number of fragments that fall into genome-wide 500-bp bins, and a Gene Score Matrix counting calculating each gene's accessibility score based on tile distance, gene size, and Tn5 insertions,

and normalising these scores across all genes. An ArchR project was subsequently created using the Arrow file for downstream analysis. We used the gene score in the *Gene Score Matrix* to infer the gene activity. Unsupervised clustering of the single-cell EpiLC data was modified from the previously described method[29]. Briefly, a selected panel of marker genes for ESC and three germ layers were obtained from previous research[27–29]. We calculated the standard deviation of gene score across all single cells for each of the four lineage types − ESC, endoderm (Endo), mesoderm (Meso), and ectoderm (Ecto). We then identified the top 50 genes with the highest standard deviation as lineage-specific markers for each cell type. To perform lineage scoring, we normalised the gene scores of these marker genes for each cell, thereby mitigating the impact of differential gene accessibility levels on scoring. For each cell, we computed the average normalised gene score of its lineage markers to derive its lineage score. Unsupervised clustering using the ward.D method was performed to generate the heatmap that depicted the transient cell states characterised by the lineage state.

### Dimensionality reduction, clustering analysis for human cell lines and PBMCs

Two additional previously published PBMC datasets that were profiled by 10X Chromium v2.1 and s3-seq[16] were integrated for comprehensive analysis. The fragment files were retrieved from NCBI GEO under the accessions GSM7102949 and GSM7102984 for 10X and s3, respectively. We used ArchR to generate Arrow files and created the ArchR project from corresponding fragment files. Iterative latent semantic indexing was performed using ArchR's function addIterativeLSI to reduce dimensions, and the Harmony[34] algorithm was used to correct different technologies' batch effect using the *addHarmony* function. Cells were clustered using *addClusters* (resolution = 0.1 for cell lines, resolution = 0.8 for PBMCs) using Seurat's FindClusters method and then embedded using UMAP by the *addUMAP* function. For PBMCs, the marker genes were identified by the *getMarkerFeatures* with GeneScoreMatrix calculated by ArchR. The MAGIC algorithm[55] was used by applying *addImputedWieghts* to impute gene scores by smoothing signals across neighbouring cells, and was used to visualise selected lineage marker genes' gene scores overlayed on the UMAP embedding. Cell identities of PBMC subset were annotated by constrained cross-platform linkage of scATAC-seq cells with scRNA-seq cells with ArchR's *addGeneIntegrationMatrix* using firstly with scRNA-seq dataset of hematopoietic differentiation[56]. By integrating the results with marker genes, the cell clusters were annotated and merged.

### Marker peaks identification and marker motif analysis using ArchR

The marker peak identification and differential motif analysis were performed by ArchR and chromVAR. For pseudo-bulk replicates, the chromatin-accessible peaks set was created using *addGroupCoverages*. Peaks were then called using the *addReproduciblePeakSet* by MACS2 for each identified cell type, and *addPeakMatrix* was used to append the count matrix of the combined peak set to the Arrow file. Differentially accessible regions(noted as marker peaks) were calculated using the *getMarkerFeatures* function, with the Wilcoxon rank-sum test chosen as the test method and plotted with *markerHeatmap* using a cut-off of FDR ≤ 0.1, $\log_2$ fold change > 1. Motif annotations were first assigned to the marker peak set using the *addMotifAnnotations* function, followed by motif enrichment analysis in marker peaks with *peakAnnoEnrichment*. To evaluate TF activity at the single-cell level, chromVAR was applied using motif annotations as a reference. The ArchR's *addBgdPeaks* function was employed to add background peaks, accounting for GC-content and fragment count similarities across samples based on Mahalanobis distance. The per-cell motif deviation scores were then computed across annotated motifs using the *addDeviationsMatrix* function, utilising the enriched marker peaks for respective clusters to build the motif deviation matrix. The top motif deviation matrix was computed using *getVarDeviations*, which is based on the chromVAR[23] algorithm. The top variable motifs were plotted with *plotVarDev*, and TF footprintings were plotted using *getPositions* and *plotFootprints* and normalised by dividing the footprinting signal by the Tn5 bias signals.

### Statistical analysis

All statistical analyses were performed using GraphPad Prism (version 9.0) and R (version 4.3.3). For pairwise comparisons, the two-sided Wilcoxon signed-rank test was employed. For multiple comparisons, *P*-values were adjusted using the Benjamini–Hochberg method to control the false discovery rate. Specific statistical tests and significance levels are detailed in the respective figure legends.

### Reporting summary

Further information on research design is available in the Nature Portfolio Reporting Summary linked to this article.

## Data availability

The datasets supporting the conclusions of this article are available in the NCBI Sequence Read Archive with accession number PRJNA1073020; SRR32538998 (IT-scATAC-seq of HEK293T cell line), SRR32538997 (IT-scATAC-seq of K562 cell line), SRR32538996 (IT-scATAC-seq of H1 cell line), SRR27862248 (IT-scATAC-seq of mixed species using HEK293T and NIH/3T3), SRR28081828 (IT-scATAC-seq of mouse ESCs differentiation), and SRR28081827 (IT-scATAC-seq of human PBMCs); GSM2970932 (sci-ATAC of GM12878 and HL-60); GSM5343842 (HyDrop-ATAC of mixture of human MCF-7 cells and mouse melanoma cells. Other datasets used are available from [https://bis.zju.edu.cn/chatac/] (fragment file of CH-ATAC of HEK293T and 3T3 cells), [https://github.com/dbrg77/plate_scATAC-seq] (quality control metrics of plate-based and C1-based scATAC-seq), [https://www.10xgenomics.com/datasets] (quality control metrics of 10X Next GEM v1.0 of mixture of GM12878 and A20 cells). Source data are provided with this paper.

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

## Acknowledgements

We thank the workshop on single-cell omics in 2019 provided by Professor He Aibin's lab at Peking University. We thank the HKU core facility

for the Flow Cytometry and Echo® 550 Liquid Handler. This work was supported by the Innovation and Technology Commission of Hong Kong grant InnoHealth@HK, Theme-based Research Scheme (T13-602/21N), Guangdong High-level Hospital Construction Project (KJ012019517), Guangdong Provincial People's Hospital Foundation (KY012021405), Science, Technology and Innovation Commission of Shenzhen Municipality (JCYJ20210324, 114408024 to Z.Z.), Guangdong-Dongguan Joint Research Scheme Guangdong-Hong Kong-Macau Program 2021B1515130004.

## Author contributions

W.J. and Z.Z. conceived the project. W.J., J.M. and L.R. designed the experiments. W.J. did the experiments with help from T.L. L.R. did the FANS part. J.M. performed the bioinformatic analysis with help from S.H. G.J. helped with data discussion. W.J. and J.M. interpreted the data and prepared the manuscript with comments and inputs from all authors. Z.Z applied for funding for this project.

## Competing interests

Z.Z., W.J., J.M., and L.R. have filed a patent application (PCT/CN2024/106967) related to this work. The remaining Authors declare no competing interests.
