## [Transparent Peer Review file · Nature Communications]

Semi-automated IT-scATAC-seq profiles cell-specific chromatin accessibility in differentiation and peripheral blood populations

Corresponding Author: Professor Zhongjun Zhou

Version 0:

Reviewer comments:

Reviewer #1

(Remarks to the Author)

Jin et al. propose a novel method for scATAC-seq that minimizes cost without compromising quality. Their method, IT-scATAC-seq, utilizes parallel bulk tagmentation to minimize benchtop variation and reduce pipetting and labor time. To test their technique, the authors compared IT-scATAC-seq libraries to libraries from other established scATAC-seq methods. While the sequencing depth was lower in IT-scATAC-seq libraries, they still had high quality, as seen by their high TSS scores, low mitochondrial DNA, and many unique fragments per cell. IT-scATAC-seq could accurately distinguish between different species, cell lines, and cell heterogeneity in human PBMCs.

Using their novel technique, Jin et al. identified cell-type-specific alterations in the epigenetic landscape of HGPS patients when compared to healthy controls. By performing KEGG on loss/gained peaks, they found enrichment in cellular senescence and immune response pathways. Lastly, they identified differentially regulated regions in HGPS PBMCs that correspond with aging, namely PRMT6.

In figure 2b, they mention a higher number of unique fragments per cell in IT-scATAC-seq libraries. It's a bit difficult to see that in the figure, especially compared to the plate method. Are there any sort of statistics/number?

In figure 2d, they show a high FRiP score in IT-scATAC-seq when compared to other methods. Is it possible that the lower sequencing depth is able to artificially inflate the FRiP score?

Since it is a novel method, it seems to be necessary to cross-validate IT-scATAC-seq using CHIP for specific markers and analyze the overlap?

They tested their method across species and cell types. Is it possible to perform a time course experiment (in response to differentiation, for example) to determine if pm IT-scATAC-seq is able to capture changes in chromatin availability across biological events, especially during differentiation or development?

I believe the work will be of significance to the field. The authors propose a novel way to perform scATAC-seq that reduces cost and makes the method more accessible to a broader array of labs. Libraries developed from IT-scATAC-seq were able to withstand comparison with established scATAC-seq methods in literature.

Reviewer #2

(Remarks to the Author)

In this study, authors presented a novel scATAC-seq methodology, the IT-scATAC-seq, ensuring that this new technology provides a scalable, semi-automated workflow capable of preparing libraries for thousands of cells within a single day. They applied this technique to blood cells from healthy donors, identifying the distinct cell populations based on chromatin accessibility profiles. A comparison between healthy individuals and patients with premature aging revealed significant chromatin accessibility loss across various immune cell types. By integrating this data with lamina-associating domains (LADs), regions of repressive heterochromatin affected in progeria, they identified altered chromatin accessibility in several aging-related genes, including YTHDC1, FXR1, LRRN3, and PRMT6. The decrease in PRMT6 levels was experimentally analyzed in another cell model.

Chromatin remodelling is one of the hallmarks of the pathological premature aging and the identification/characterization of molecular mechanisms driving this aberrant process is of interest for the scientific community. Understanding gene regulation at the single-cell level is crucial for studying cellular heterogeneity in complex biological systems, thus the adopted approach is very interesting.

However, a significant limitation of this study is the use of data from a single HGPS patient and two pooled controls, with only one technical replicate. Given the variability in individual epigenetic profiles, this approach is problematic. I also find the use and comparison of distinct cell sources to be inappropriate, leading to several instances of overinterpretation.

Therefore, I cannot support this manuscript's publication in Nature Communications.

Major comments:

1. Figure 1: The differences and advantages of the novel scATAC-seq set up are not clearly explained. The provided scheme and supplementary figures are insufficient. It is also unclear how this method differs from the approach presented by "Cusanovich et al. Cell 2018".
2. Figure 2: The comparisons presented in the text are overly speculative. The authors stated: "Despite lower sequencing depths (median duplication rate 54-57% for IT-scATAC-seq versus over 95% for plate-based and C1scATAC-seq) (Fig. 2a)", however the duplication rate is not lower than all the presented protocols; or: "Additionally, IT-scATAC-seq produced higher median TSS enrichment scores (Fig. 2e)", but this is true only for the HEK293T whose result is similar to the one obtained by CH-based method.
3. Figure 3: In the IT scATAC-seq analysis of PBMCs shown in Figure 3b, the cell populations are not well-separated, particularly the CD4 subtypes. Does this ATAC-based cell characterization truly reflect genome function? The authors had to rely on scRNA-seq of PBMCs to refine cell identification. Therefore, what is the advantage of using scATAC-seq over scRNA-seq in this context?
4. Figure 3: The authors compared their IT-scATAC-seq with two available PBMC scATAC-seq datasets generated using different methods (WT 10X and WT s3). However, the rationale for selecting only these two controls is unclear, especially given the eight different scATAC-seq protocols mentioned in De Rop et al., Nat Biotech 2023, which the authors cite.
5. Figure 4: The interpretation of the differences between the controls and the single HGPS sample is challenging to follow. Initially, the authors mention an overall decrease in chromatin accessibility but then shift focus to upregulated peaks. However, several downregulated peaks are also linked to T lymphocyte activation, such as the mTOR-signaling pathway. How do the authors justify this discrepancy? They wrote: "Functional analysis of down-regulated peak-associated genes revealed enrichment in immune response pathways, including cell adhesion, lymphocyte differentiation, chemokine, signalling, cytokine production, and infection-related pathways". However, cytokine production, signaling, lymphocyte differentiation are terms characterizing the immune system activation. Notably, one of the KEGG terms associated with the downregulated peaks is "cellular senescence". They concluded: "cellular senescence pathways were significantly enriched across all cell types aligning with accelerated immunosenescence in HGPS observed in previous studies", however, this conclusion appears to be an overinterpretation based on the GO analysis in Figure 4b.
6. Figure 4: Are the upregulated ATAC-seq peaks in HGPS derived PBMC indicative of a specific epigenetic chromatin state, or are they also linked to active transcription? Expression analysis is essential for any interpretation of function and dysfunction.
7. Figure 5: The authors chose to intersect the differentially accessible peaks with LAD mapped in HeLa and HAP-1 cells. While most of the LADs are conserved, they opted for a cancer cell line, which likely has an altered chromatin structure, while HAP-1 is a haploid cell line. Why didn't they consider using LADs from a human primary cell line, such as fibroblasts? In the paper cited by the authors, Kohler et al. (Genome Med. 2020) present LAD data for both wild-type and HGPS fibroblasts. Why did the authors not consider the HGPS LADs instead of those from tumor cells? Additionally, Kohler et al. also provide ATAC-seq data; how are the pathological euchromatin changes conserved between fibroblasts and PBMCs? Are they conserved or cell specific?
8. Figure 5: the authors stated Page 10 lane 274: "Notably, most immune cell types showed a marked loss of peaks in both A-LADs and B-LADs, but CD4 naive T cells exhibited more gained A-LADs-associated peaks (393 gained vs. 297 lost) and comparable alterations in B-LADs-associated peaks (1,146 gained vs. 1,194 lost), suggesting cell-type-specific resilience to HGPS-induced chromatin alterations", however, this is an overinterpretation given the lack of Lamin ChIP-seq data in these cells.
9. Figure 6: How did the authors select the set of genes shown in Figure 6a? Are all the genes with reduced chromatin accessibility in LAD across various cell populations included, or did they choose only the most interesting ones?
10. Figure 6: With a set of interesting genes, it would be beneficial to include their expression profiles, even in a bulk population. RNA-seq data for HGPS patients is available, although not specifically for PBMCs.
11. Figure 6: PRMT6 is located within a Lamin B domain, whereas HGPS results from a mutation in Lamin A. Is the gene situated within a domain shared by both Lamin A and Lamin B? If not, how do the authors comment this detail?
12. Figure 6: The siRNA data is unconvincing to address the role of PRMT6 in premature senescence. As expected, reduced PRMT6 determines cellular senescence as previously described, but its expression is crucial in the HGPS? Why didn't the authors attempt to overexpress PRMT6 in the MSCs differentiated from HGPS patient-derived iPSCs to revert the premature senescence?

Minor comments:

- 1) In several panels, the text is too small to be readable on a printed page. Examples: Figure 1J, Extended Figure 5a, Figure 4b, figure 4c.
- 2) Page 6 lane 129 "...20 randomly selected single-cell profiles demonstrated high congruence with bulk data, with Pearson correlation coefficients ranging from 0.71 to 0.94". However, the Extended Data Fig. 5a shows a correlation ranging from

0.52 to 0.94

3) The order of protocols presented on y axis of distinct panels of Figure 2 is not the same for all, making it challenging to follow.

4) One of the claims is that the protocol can be completed in a single day; however, no information about the other protocols was provided.

5) Another assertion made by the authors is the simplicity of the protocol and the absence of overly complicated or expensive instruments. However, the automated Labcyte Echo 55 Acoustic liquid handler is quite costly and is not available in every laboratory.

Version 1:

Reviewer comments:

Reviewer #1

(Remarks to the Author)

The authors have addressed my concerns.

Reviewer #2

(Remarks to the Author)

Reviewer #2 (Remarks to the Author): In this study, authors presented a novel scATAC-seq methodology, the IT-scATAC-seq, ensuring that this new technology provides a scalable, semi-automated workflow capable of preparing libraries for thousands of cells within a single day. They applied this technique to blood cells from healthy donors, identifying the distinct cell populations based on chromatin accessibility profiles. A comparison between healthy individuals and patients with premature aging revealed significant chromatin accessibility loss across various immune cell types. By integrating this data with lamina-associating domains (LADs), regions of repressive heterochromatin affected in progeria, they identified altered chromatin accessibility in several aging-related genes, including YTHDC1, FXR1, LRRN3, and PRMT6. The decrease in PRMT6 levels was experimentally analysed in another cell model. Chromatin remodelling is one of the hallmarks of the pathological premature aging and the identification/characterization of molecular mechanisms driving this aberrant process is of interest for the scientific community. Understanding gene regulation at the single-cell level is crucial for studying cellular heterogeneity in complex biological systems, thus the adopted approach is very interesting. However, a significant limitation of this study is the use of data from a single HGPS patient and two pooled controls, with only one technical replicate. Given the variability in individual epigenetic profiles, this approach is problematic. I also find the use and comparison of distinct cell sources to be inappropriate, leading to several instances of overinterpretation.

Response: We thank you for the thoughtful assessment and acknowledge the concerns regarding sample limitations. As noted in our manuscript, the primary aim of this study is to showcase IT-scATAC-seq as a novel, scalable approach to single-cell ATAC-seq, demonstrating its potential application with PBMCs from an HGPS patient. The aim of this manuscript is to describe the development of a new technology, and its application on analysing this single HGPS sample serves as a proof of concept, focused on assessing chromatin accessibility patterns and their alignment with known aging-related pathways.

We recognise that using a single HGPS patient sample limits the ability to make broad conclusions about progeroid immunosenescence. HGPS is an exceptionally rare disease, occurring in approximately 1 in every 4 million live births worldwide. The scarcity of cases, along with significant health risks for these patients, makes sample acquisition inherently challenging, and thus, we are not able to obtain samples from more patients at this moment. To emphasise the limitation, the use of only one patient sample is stated in the revised manuscript in the abstract, main text, and especially in the discussion to clarify that our findings are not generalised to all HGPS cases. To mitigate variability, we have carefully chosen controls and reference datasets, as outlined in our responses to specific comments, ensuring robust comparisons and minimising potential biases in the analysis.

Regarding the comparison of cell sources, we apologise if any unintended ambiguity contributed to the impression of inappropriate use. Our integration of LAD-associated chromatin accessibility follows established methodologies for identifying altered regulatory regions in progeroid cells. We have clarified our rationale in the revised manuscript to prevent misinterpretation.

Additionally, in response to this concern, we have toned down the manuscript title, abstract and relevant description in the results section to avoid implying functional conclusions and to emphasise that our observations are focused on epigenetic changes.

We trust that these clarifications address your main concerns, with further specifics provided in our point-by-point responses below.

REVIEWER #2*****

Overall, I find this revision insufficient, as none of my concerns have been adequately addressed. The responses provided are more structured but largely reiterate arguments already present in the original text. Below, a detailed point-by-point commentary.

Number of HGPS patients:

In response to my request to include more HGPS patients, the authors stated that the rarity of the disease limits material

availability. They wrote in their rebuttal: “The scarcity of cases, along with significant health risks for these patients, makes sample acquisition inherently challenging, and thus, we are not able to obtain samples from more patients at this moment” However, the Progeria Research Foundation (<https://www.progeriaresearch.org/>) offers several cell lines for scientific purposes. These include multiple lymphoblast cell lines which, although not identical to PBMCs, could be used for B cell validation experiments.

Comparison with cancer cells:

This point remains unaddressed. Despite the availability of several datasets on LADs in healthy cells, the authors did not use other LAD datasets. They wrote: “Our integration of LAD-associated chromatin accessibility follows established methodologies for identifying altered regulatory regions in progeroid cells” However, this response does not address my concern, which was about the rationale behind the comparison, not the methodology used in the analysis.

The progeria focus:

The central theme of the paper remains predominantly centered on progeria, with four out of seven figures dedicated to this topic. In my view, the data provided does not sufficiently support this focus. If the intent of this work is to promote the technology, I would be open to reconsider my position. However, the sections addressing the HGPS sample should be presented as descriptive, with explicit explanations provided in the text.

1. Figure 1: The differences and advantages of the novel scATAC-seq set up are not clearly explained. The provided scheme and supplementary figures are insufficient. It is also unclear how this method differs from the approach presented by “Cusanovich et al. Cell 2018”.

Response: We thank your feedback and would like to clarify that the differences and advantages of IT-scATAC-seq are detailed throughout the manuscript in multiple sections.

In the Introduction part, we discuss the limitations of current scATAC-seq technologies, particularly in terms of the challenges in costs, sensitivity and scalability. IT-scATAC-seq was specifically designed to overcome these limitations and disadvantages, providing a scalable, cost-effective, and adaptable solution for various research settings.

The Results section includes a comparison with conventional scATAC-seq methods in the subsection titled “Comparing IT-scATAC to other scATAC-seq methods.” This section highlights the protocol’s advantages in terms of library complexity, TSS enrichment, and cost. IT-scATAC-seq enables high-throughput processing of up to 10,000 cells per day while minimising reagent costs, achieving up to a 100-fold reduction in per-cell costs compared to traditional plate-based methods. Additionally, the protocol does not require any equipment specialised for single-cell libraries (in contrast to 10X and Fluidigm C1) and can be performed in a wide range of laboratories.

In the Discussion, we again summarise IT-scATAC-seq’s streamlined workflow, which leverages indexed Tn5 bulk tagmentation to reduce labour and handling time. While we do use the Echo 550 liquid handler to further minimise manual pipetting, this instrument is not essential to our protocol. IT-scATAC-seq is fully adaptable and remains efficient and cost-effective even with standard laboratory equipment.

We also summarise its advantages here:

- Scalability: IT-scATAC-seq enhances the throughput, increasing cell processing capacity to match high-throughput methods like droplet-based scATAC-seq. The protocol can be scaled by modularly increasing indexed Tn5 transposomes or PCR primers, with minimal oligo synthesis needed, making it suitable for both small- and large-scale applications.
- Accuracy: By using cell sorting rather than droplet-based systems, IT-scATAC-seq minimises the doublet rate and avoids barcode misassignment issues commonly associated with droplet and combinatorial indexing methods.
- Hands-on time: IT-scATAC-seq requires less manual labour than traditional plate-based scATAC-seq. Using the Echo liquid handler further streamlines pipetting but is not essential to the workflow, allowing labs without access to this equipment still able to implement IT-scATAC-seq efficiently.
- Cost-effectiveness: IT scATAC-seq achieves up to a 100-fold reduction in per-cell cost compared to conventional scATAC-seq. The protocol’s minimised PCR volume and efficient reagent usage reduce library preparation costs to approximately 0.1 HKD (~\$0.01) per cell, a level of cost-efficiency that is readily achievable with standard lab reagents.

Regarding the difference between the two methods IT-scATAC-seq and sci-ATAC-seq (combinatorial indexing) used in “Cusanovich et al. Cell 2018”, we contrast here:

1) Indexing method:

a. Single-cell combinatorial Indexing: This method involves rounds of pooling and splitting with barcoded Tn5 transposases in wells. Nuclei are tagged in bulk in 96-well plates, pooled, and then redistributed into a second set of wells where a second barcode is introduced during PCR. This process relies on the assumption that nuclei indexed in the first round will not be pooled into the same well in the second round, but it still suffers from collisions (doublet) due to the random nature of the pool and splitting.

b. IT-scATAC-seq: We utilize limited parallel labelling for the first-round indexes, followed by flow-cytometry-based single-cell sorting. This ensures that every cell receives a unique barcode, which guarantees high accuracy and low doublet rates. This approach avoids the collisions inherent in combinatorial indexing.

2) Scale of Indexed Tn5: sci-ATAC-seq requires the assembly of $96 * N$ indexed Tn5 transposases (e.g., 384 Tn5 complexes when four 96-well plates are used in the first round). Only a few indexed Tn5 transposases are needed for the first round in IT-scATAC-seq (e.g., less than 12), significantly simplifying the process and reducing potential errors. Therefore, the handling of tagmentation in IT-scATAC-seq is much easier than in sci-ATAC-seq.

3) Collision: sci-ATAC-seq involves rounds of pooling and splitting with barcoded Tn5 transposases in wells. Nuclei are

tagged in bulk in 96-well plates, pooled, and then redistributed into a second set of wells where a second barcode is introduced during PCR. This process relies on the assumption that nuclei indexed in the first round will not be pooled into the same well in the second round, but it still suffers from collisions due to the random nature of pool and splitting. IT-scATAC-seq utilizes limited parallel labelling for the first-round indexes, followed by flow-cytometry-based single-cell sorting. This ensures that every cell receives a unique barcode, which guarantees high accuracy and low doublet rates. This approach avoids the collisions inherent in combinatorial indexing.

4) Barcode utilization: > 80% of cell barcodes in sci-ATAC-seq are unused, while all the barcodes in IT-scATAC-seq are used.

5) Customized adapters: Considering the Nextera adapter sequences occupy the Tn5 binding sequence, sci-ATAC-seq needs to design customized adapters for sequencing. We integrate the TruSeq adapters into our library to generate a three-round cell indexing strategy in IT-scATAC-seq, which not only increases the throughput flexibility but can also be multiplexed with other samples for the Illumina sequencing platform.

REVIEWER #2*****

The advantages of the technology outlined by the authors in the first part of their response were already evident from the original text. Differences from the methodology described in Cusanovich et al., Cell (2018) are now clearer; it would be helpful to summarize these distinctions in a supplementary table for the benefit of readers who are not experts in ATAC-seq.

2. Figure 2: The comparisons presented in the text are overly speculative. The authors stated: "Despite lower sequencing depths (median duplication rate 54-57% for IT-scATAC-seq versus over 95% for plate-based and C1scATAC-seq) (Fig. 2a)", however the duplication rate is not lower than all the presented protocols; or: "Additionally, IT-scATAC-seq produced higher median TSS enrichment scores (Fig. 2e)", but this is true only for the HEK293T whose result is similar to the one obtained by CH-based method.

Response: We appreciate your feedback and apologise if the presentation of the data was not as clear as intended. We thank the opportunity to clarify our presentation of the data comparing IT-scATAC-seq with other established methods. Figure 2, panels a and b, compare duplication rates and unique fragments per cell across different scATAC-seq methods, metrics that reflect library complexity and sequencing depth. Generally, higher duplication rates indicate greater sequencing depth, which in turn increases the number of unique fragments up until sequencing depth reaches saturation (Reference: PMID 37537502). By writing,

"Despite lower sequencing depths (median duplication rate 54-57% for IT-scATAC-seq versus over 95% for plate-based and C1 scATAC-seq) (Fig. 2a), IT-scATAC-seq exhibited higher or comparable library complexity, indicated by a higher number of unique fragments per cell (Fig. 2b).", we described that in our data, although the duplication rates (and thus sequencing depths) for plate-based and C1 methods are much higher than those for IT-scATAC-seq across all cell lines tested, IT-scATAC-seq demonstrates higher (in the case of HEK293T) or similar performance (in the case of K562 and H1) in terms of unique fragments, suggesting that under a lower sequencing depth, we can achieve comparable or higher library complexity.

We have included detailed metrics along with our response comparing the performance of plate-based methods and IT-scATAC-seq as supplementary materials to provide a clear basis for these conclusions.

For the TSS enrichment score, we thank and acknowledge the reviewer's point and have revised the text to state:

"Additionally, IT-scATAC-seq produced higher or similar median TSS enrichment scores compared with existing methods (Fig. 2e)."

REVIEWER #2*****

I appreciated the revision of the text

3. Figure 3: In the IT scATAC-seq analysis of PBMCs shown in Figure 3b, the cell populations are not well-separated, particularly the CD4 subtypes. Does this ATAC-based cell characterization truly reflect genome function? The authors had to rely on scRNA-seq of PBMCs to refine cell identification. Therefore, what is the advantage of using scATAC-seq over scRNA-seq in this context?

Response: Thank you for your question regarding scATAC-seq.

1) Reflection of genome function: ATAC-seq is a well-established method for assessing chromatin accessibility, which is a crucial epigenetic aspect of genome function. This technique has been widely used to infer regulatory landscapes behind gene expression dynamically and are not directly observable through transcriptome. The scATAC-seq examples include studies by Buenrostro et al. Nature. (2015) (PMID: 26083756), Zhang et al. Cell. (2021) (PMID: 34774128), Corces et al. Nature Genetics. (2020) (PMID:333106633), and Becker et al. Nature Genetics (2022) (PIMD:35726067), where scATAC-seq effectively mapped enhancer and promoter activities across different cell types and conditions like cancer and Alzheimer's.

2) Challenges in cell population separation: The separation of closely related cell subtypes, such as CD4 T cell populations, using scATAC-seq can indeed be challenging, a limitation that is well-observed in the scATAC-seq studies. This difficulty primarily stems from the nature of chromatin accessibility data, which may not always provide as distinct markers for cell type differentiation as transcriptomic data. This has been demonstrated by De Rop et al. (2024, PMID: 37537502), in their systematic benchmarking of scATAC-seq technologies using human PBMCs; similarly when Mezger et al. (2018, PMID: 30194434) developed a scATAC-seq technology and applied it to human PBMCs, as well as many studies analyzing aging, aging-related conditions, and infections such as COVID-19 using human PBMCs have also demonstrated these limitations (e.g., PMID: 32780218, 37081034, PMID: 37095391). In this study, the annotation between naive and memory CD4 T cells was based on integrated scRNA-seq data and specific cell markers.

3) Integration with scRNA-seq: We did not solely rely on scRNA-seq for cell population annotation. Integrating scRNA-seq

data is one common approach to enhance the annotation of cell types identified by scATAC-seq, utilizing gene expression profiles to corroborate and refine chromatin accessibility-based predictions (PMID:34725479,33106633). Manual annotation based on known marker genes is also viable and is employed where appropriate, but integration with scRNA-seq can provide a more robust and detailed cell type classification, particularly for closely related subtypes. We also showed the enhanced accessibility of cell type-specific markers of different populations in Figure 3c.

4) Advantages of scATAC-seq over scRNA-seq: While scRNA-seq is a powerful method for capturing gene expression profiles, scATAC-seq provides unique and complementary information by directly measuring chromatin accessibility, thereby revealing the regulatory landscape that drives gene expression changes. Previous works have demonstrated the epigenetic deregulation in HGPS cells, but no epigenetic results have ever been reported for patient's blood cells. In the context of our study, scATAC-seq allows us to pinpoint active regulatory elements, such as enhancers and promoters, and to identify specific chromatin accessibility patterns associated with HGPS that influence gene regulation beyond the transcriptional layer. For instance, we observed LAD-associated changes in chromatin accessibility—an indication of nuclear lamina disruptions common in progeroid syndromes like HGPS—that are inherently epigenetic and not detectable by transcriptional profiling alone. This does not imply that scRNA-seq is less valuable in this context; indeed, it remains a powerful tool for disease research. However, the focus of our current work is on demonstrating the broad utility of our new scATAC-seq technology and providing novel insights into the epigenetic alterations associated with HGPS, which is beyond what scRNA-seq can reveal. We agree that although exploring the expression patterns in HGPS is beyond the scope of this study, such analysis would be a valuable direction for future research.

REVIEWER #2*****

While I appreciate the detailed explanations provided regarding the ATAC-seq, I still think that, at present, this technology cannot provide functional data without being complemented by RNA-seq. Furthermore, while the authors emphasized that the primary focus of the paper is on describing a new technology, I found that is still focused on progeria.

Additionally, given the inherent variability of epigenetic signatures, the findings derived from a single patient, without validation in additional samples, lack sufficient scientific rigor and reliability.

4. Figure 3: The authors compared their IT-scATAC-seq with two available PBMC scATAC-seq datasets generated using different methods (WT 10X and WT s3). However, the rationale for selecting only these two controls is unclear, especially given the eight different scATAC-seq protocols mentioned in De Rop et al., Nat Biotech 2023, which the authors cite.

Response: Thank you for your question regarding our selection of control datasets in Figure 3, and I apologise for not being clearer in the explanation of the rationale. In our comparison, we opted to use datasets from the study by De Rop et al., Nat Biotech 2023, which extensively characterised PBMCs using eight different scATAC-seq methods: 10x Genomics scATAC-seq (v1, v1.1, v2, multiome and mitochondrial scATAC (mtscATAC)) as well as Bio-Rad ddSEQ, HyDrop and s3-ATAC. The same PBMC cell source (pooled two healthy samples) was used across eight different methods: "Cryopreserved human PBMCs from one male donor and one female donor were purchased from AllCells and distributed across institutes to generate the following samples.....".

The eight methods in the report by De Rop et al. are primarily categorised into two groups: droplet-based (10X v1, v1.1, v2, ddSEQ, HyDrop) and sci-based protocols (s3) (10X multiome and mtscATAC are multi-omics profiling, and mitochondrial profiling is a variant of 10X that profiles other modalities rather than chromatin accessibility). We selected one representative dataset from each category—10X v1.1 for droplet-based and s3 for sci-based—to provide a broad perspective on the most prevalent technologies in the field. By incorporating these datasets, we aimed to minimise discrepancies arising from both distinct technological approaches and potential biological variability. In other words, using one dataset from our own research as a biological WT reference, alongside the two additional WT datasets from De Rop et al. processed by different technologies, ensures our comparison between HGPS and WT PBMCs is robust, minimising the impact of both technological and biological variations.

REVIEWER #2*****

This explanation could be incorporated into the Methods section to assist the reader.

5. Figure 4: The interpretation of the differences between the controls and the single HGPS sample is challenging to follow. Initially, the authors mention an overall decrease in chromatin accessibility but then shift focus to upregulated peaks.

However, several downregulated peaks are also linked to T lymphocyte activation, such as the mTOR-signalling pathway. How do the authors justify this discrepancy? They wrote: "Functional analysis of down-regulated peak-associated genes revealed enrichment in immune response pathways, including cell adhesion, lymphocyte differentiation, chemokine, signalling, cytokine production, and infection-related pathways". However, cytokine production, signaling, lymphocyte differentiation are terms characterizing the immune system activation. Notably, one of the KEGG terms associated with the downregulated peaks is "cellular senescence". They concluded: "cellular senescence pathways were significantly enriched across all cell types aligning with accelerated immunosenescence in HGPS observed in previous studies", however, this conclusion appears to be an overinterpretation based on the GO analysis in Figure 4b.

Response: We thank you for your detailed comments and would like to clarify several aspects regarding our analysis and interpretation of chromatin accessibility changes in HGPS PBMCs.

Our analysis identified the observed overall decrease in chromatin accessibility across all immune cell populations analysed in HGPS (Figure 5a). Contrary to "shift focus to upregulated peaks", our study examines both regions with increased accessibility (up-regulated peaks) and regions with decreased accessibility (down-regulated peaks) (Figure 5b-c, Extended Data Fig.8), presenting a balanced view of chromatin state alterations in HGPS PBMCs.

For regions with decreased chromatin accessibility (down-regulated peaks, Figure 5c), we conducted an enrichment

analysis to identify the pathways and processes potentially affected by LMNA mutation in HGPS patient's PBMCs. Genes associated with these downregulated peaks (2.5kb ± of TSS region) were found to be enriched in pathways related to cell adhesion, lymphocyte differentiation, chemokine signalling, cytokine production, and infection responses. This does not indicate or imply immune activation in these populations. Rather, it suggests that genes (could be pro- or anti-cell adhesion, lymphocyte differentiation, infection, cytokine production) that are involved in these immune-related processes exhibited reduced accessibility, potentially affecting the regulation of these pathways. Notably, previous studies on lamin A/C-related immune cell function support this interpretation, showing that lamin A/C mutations or deficiencies impact immune regulation, leading to altered immune signaling, activation, and aging-related immune deficits, which has been reviewed by Saez et al. (2020, PMID: 32854281).

Regarding the KEGG "Cellular Senescence" pathway (hsa04218), we observed that this pathway was enriched among down-regulated peaks across all analyzed cell populations. It is important to note that hsa04218 includes both pro-senescent and anti-senescent genes (<https://www.genome.jp/pathway/hsa04218>). Thus, our analysis only shows that the accessibility (expression regulation) of cellular senescence-related genes is reduced in HGPS PBMCs, which are implicated in HGPS-linked aging phenotypes. This finding does not imply that "cellular senescence" itself is downregulated in these cells, nor does it suggest a causal relationship between decreased accessibility of these genes and senescence phenotypes. We believe this provides valuable insight into the chromatin state changes associated with accelerated aging in HGPS without overinterpreting the data.

Together, our findings reveal a pattern of chromatin accessibility changes in HGPS PBMCs that aligns with documented immunosenescence features, including reduced accessibility in genes linked to immune response pathways and cellular senescence. To avoid inferring any functional implications of these analyses, we have also revised the description of the relevant parts in the manuscript to emphasise the findings that were made for the epigenetic changes. We hope this clarifies the confusion and addresses your concerns about the data interpretation.

REVIEWER #2*****

This analysis has not been sufficiently improved in the revised version. As the data is derived from lymphocytes, it is unsurprising that most terms pertain to the immune system. However, unless the authors examine single genes to justify the presence of the same or similar terms in both upregulated and downregulated categories, the conclusions will continue to appear speculative. For instance, while mTOR signaling is detected in both upregulated and downregulated peaks in CD8 cells, the authors only discuss its involvement in the upregulated peaks.

In the GO Biological Process (BP) analysis, the authors stated for upregulated peaks: "Similarly, GO analysis for biological processes (BP) showed gene activity enrichment in lymphocyte differentiation, immune response-activating signalling pathways, and cell adhesion (Fig. 5b)" However, the same categories are also present in the downregulated peaks. This overlap is misleading and should be clearly addressed in the text.

On the other hand, the term "cellular senescence" appears only in downregulated peaks. Regarding this, the authors wrote: "Importantly, cellular senescence pathways were significantly enriched across all cell types aligning with accelerated immunosenescence in HGPS observed in previous studies (Fig. 5c)" citing the Gonzalo paper (Cell Reports 2018).

However, this cited study, which analyzed experiments conducted on four distinct HGPS-derived fibroblast cell lines, reported an upregulation of immune gene expression specifically in HGPS (measured at the RNA level) and highlighted differences at the single-gene level in the GO category. While the overarching conclusion of that paper links immune-related pathway upregulation to cellular senescence, it did not report the "cellular senescence" among GO terms.

The authors should analyze the terms within the GO class and provide single-gene level tracks to improve the paper's quality and reliability. Given that cellular senescence is observed in the HGPS downregulated peaks, I would expect preferential inclusion of gero-protective genes.

Again, with only a single replicate, it becomes challenging to identify reliable pathways through intersection and selection. This limitation significantly impacts the robustness of the findings.

6. Figure 4: Are the upregulated ATAC-seq peaks in HGPS-derived PBMC indicative of a specific epigenetic chromatin state, or are they also linked to active transcription? Expression analysis is essential for any interpretation of function and dysfunction.

Response: We appreciate your interest in the relationship between ATAC-seq peaks and active transcription.

Upregulated ATAC-seq peaks reflect regions of increased chromatin accessibility at regulatory elements such as TSS, enhancers, and promoters, which suggest the potential for transcriptional regulation but do not directly measure gene expression.

Importantly, ATAC-seq and scRNA-seq capture distinct regulatory and expression-level information, each providing unique insights into cellular states. In this study, the primary aim is to use IT-scATAC-seq to explore epigenetic chromatin states in HGPS-derived PBMCs, focusing specifically on chromatin accessibility as an indicator of regulatory potential rather than gene expression.

To gain a complete understanding of cellular function and dysfunction (which we neither covered nor intended to cover in this study), expression analysis alone is often insufficient. scRNA-seq, for example, is limited by its reliance on polyadenylated RNA capture, which restricts transcriptomic coverage and omits certain RNA species. Furthermore, gene expression levels do not reliably predict protein abundance, as post-transcriptional modifications and other regulatory mechanisms play crucial roles in shaping cellular functions.

We recognise the value of a multimodal approach to achieve a fuller understanding of chromatin state and transcriptional activity and plan to explore this in future studies as techniques evolve further.

REVIEWER #2*****

This response does not address my question. I inquired whether the intersection of upregulated peaks with specific histone marks reveals a preferential chromatin state affected by the disease. For instance, are bivalent promoters marked by

H3K4me3/H3K27me3 particularly affected in HGPS, as previously reported (Della Valle et al., *Sci Transl Med*, 2022; Sebestyen et al., *Nat Commun*, 2020; Salvarani et al., *Nat Commun*, 2019; Briand et al., *Hum Mol Genet*, 2018)?

7. Figure 5: The authors chose to intersect the differentially accessible peaks with LAD mapped in HeLa and HAP-1 cells. While most of the LADs are conserved, they opted for a cancer cell line, which likely has an altered chromatin structure, while HAP-1 is a haploid cell line. Why didn't they consider using LADs from a human primary cell line, such as fibroblasts? In the paper cited by the authors, Kohler et al. (*Genome Med.* 2020) present LAD data for both wild-type and HGPS fibroblasts. Why did the authors not consider the HGPS LADs instead of those from tumor cells? Additionally, Kohler et al. also provide ATAC-seq data; how are the pathological euchromatin changes conserved between fibroblasts and PBMCs? Are they conserved or cell specific?

Response: We thank you for your question regarding LAD dataset selection.

LAD data from primary fibroblasts are indeed scarce due to the limited availability of high-quality lamin A/C ChIP-seq data from these cells. Yet, LAD regions are known to be relatively conserved across cell types Kind et al. *Cell*. 2015 (PMID: 26365489). It is worth noting that in the cited study by Kohler et al. (*Genome Med.* 2020), they did not present LAD data for wildtype nor HGPS fibroblast; instead, they also relied on LAD data from HeLa cells rather than directly profiling LADs in fibroblasts, due to the high conservation of LAD domains across cell types.

"For the comparison of LAD- and "solo-WCGW" CpG probe methylation levels between HGPS and control samples, we used previously published locations of lamin A LADs (PMID: 25602132), lamin B LADs (PMID: 18463634) ..."

Following what has been used by Kohler et al., we used the same HeLa-derived LAD dataset for A-LADs. For B-LADs, we used the dataset generated from K562 cells (PMID: 32893442).

In our analysis, we observed cell-type-specific changes in chromatin accessibility across different immune cell types within HGPS-derived PBMCs, revealing euchromatin alteration signatures unique to each subtype. As for comparison with fibroblast data, Köhler et al. reported relatively fewer differentially accessible peaks in HGPS fibroblasts than we have identified in PBMCs. It is difficult to determine whether these differences are due to cell type or variations in analytical methods. Interestingly, FOS2 and BACH2 (among other AP1 family TFs), enriched in differentially accessible regions in HGPS fibroblasts identified in Köhler et al. study, were also found enriched in CD4 Memory cells of in the HGPS patient (Supplementary Figure 9b).

REVIEWER #2*****

The authors wrote in their rebuttal: "LAD data from primary fibroblasts are indeed scarce due to the limited availability of high-quality lamin A/C ChIP-seq data from these cells". LAD data from primary fibroblasts are not scarce. Lamin-associated domains (LADs) have been mapped in human fibroblasts in several studies, including McCord et al. (*Genome Research*, 2013), Sadaie et al. (*Genes & Development*, 2013), Dou et al. (*Nature*, 2015), and Paulsen et al. (*Genome Biology*, 2017). For example, in the McCord paper, the authors used two different anti-lamin A/C antibodies (MAB3211 and N18) to map LADs in two biological replicates of HGPS and normal fibroblast cells. Additionally, LAD datasets are available for Jurkat T cells (Robson et al., *Genome Research*, 2017), which are more similar to the model system used in the present study.

The authors in their rebuttal have incorrectly cited Kind et al. (*Genome Research*, 2012) as the source for the conservation theory of LADs across cell types. This finding should instead be attributed to Meuleman et al. (*Genome Research*, 2012). Kind et al. (same lab, but a later study) performed single-cell analyses, reporting significant variability in LAD contacts even within the same cell population, with LADs covering the 35% of the genome and only 15% of LADs classified as structural and present in all cells. This distinction underscores the current understanding of LADs: there are constitutive LADs (cLADs), which are cell-type invariant, and facultative LADs (fLADs), which are cell-type specific and play crucial roles in cell identity. This concept is summarized by Bas van Steensel in his review on *Cell* (2017), where he estimated that fLADs constitute at least half of all detected LADs.

Another important information is that LAD remodeling is a hallmark of differentiation and disease, as previously shown by Reddy et al. (*Nature*, 2008) and extensively characterized by the Collas lab (Lund et al., *Genome Research*, 2013; Ronningen et al., *Genome Research*, 2015; Madsen-Osterbye et al., *Genome Biology*, 2022; Benarroch et al., *Cells*, 2023). Cancer cells, too, exhibit LAD remodeling, as demonstrated by Lenain et al. (*Genome Research*, 2017) and Ji et al. (*Biochim Biophys Acta Gene Regul Mech.*, 2020). Thus, due to the extensive alterations in LAD organization seen in cancer cells, they are not suitable as a reference dataset for LAD analysis in healthy cells.

8. Figure 5: the authors stated Page 10 lane 274: "Notably, most immune cell types showed a marked loss of peaks in both A-LADs and B-LADs, but CD4 naive T cells exhibited more gained A-LADs-associated peaks (393 gained vs. 297 lost) and comparable alterations in B-LADs-associated peaks (1,146 gained vs. 1,194 lost), suggesting cell-type-specific resilience to HGPS-induced chromatin alterations", however, this is an overinterpretation given the lack of Lamin ChIP-seq data in these cells.

Response: We thank you for your comments and acknowledge that Lamin A/C ChIP-seq data would provide a more direct assessment of LAD structural changes in HGPS. However, our study's objective is to investigate chromatin accessibility changes in specific cell types in HGPS PBMCs, with LAD regions (which are conserved among human cell types) of particular interest, rather than to examine broader structural changes in LAD domains themselves.

Our statement regarding "resilience" in CD4 naive T cells reflects the observation that, unlike other immune cell types, these cells exhibited a unique pattern of peak gains in A-LADs (393 gained vs. 297 lost). This pattern suggests a distinct chromatin accessibility change pattern within the LADs region of CD4 naive T cells under HGPS conditions rather than implying structural resilience in LAD regions. We have tone-downed the text to clarify that this interpretation is based on observed chromatin accessibility patterns within LAD-associated regions relative to other types of cells and does not imply direct structural changes in LADs of patient's cell populations without ChIP-seq confirmation.

REVIEWER #2*****

I believe there is still an element of overspeculation in this statement. When you assert that "...CD4 naive T cells exhibited significantly more gained A-LADs-associated peaks..." it gives the impression that these peaks are located within LADs. However, without Lamin ChIP-seq data for these cells, such a claim cannot be supported. On the other hand, this statement does not contribute meaningfully to the paper's core message and, in my opinion, can be omitted.

9. Figure 6: How did the authors select the set of genes shown in Figure 6a? Are all the genes with reduced chromatin accessibility in LAD across various cell populations included, or did they choose only the most interesting ones?
Response: We thank you for the question regarding the selection criteria for the set of genes shown in Figure 6a. The genes depicted are representative of regions with altered chromatin accessibility in LADs across various cell populations, specifically those that overlap with findings from a previously published aging-related study. As outlined in the manuscript, we focused on genes that exhibited significant loss of chromatin accessibility in LADs ($p < 0.05$) and correlated with decreased expression associated with chronological aging. This targeted approach ensured that we included genes with known relevance to both HGPS and aging.

To clarify, the complete list of overlapped genes with significantly altered accessibility is detailed in Supplementary Data 5 and 6. Among them, we selected a subset that best represented the key findings of our study.

The above has also been noted in the original manuscript: "To identify regions in HGPS PBMCs that are differentially regulated and associated with aging, we compared chromatin accessibility changes in HGPS samples with genes known to be affected by chronological aging. We focused on genes differentially expressed with chronological age with a negative Z-score, indicating decreased expression with aging, and mapped these regions where LAD-associated chromatin accessibility was significantly lost in HGPS compared to healthy controls".

REVIEWER #2*****

In their response, the authors kindly copied and pasted the same explanation from the text. I apologize if my initial request was not sufficiently clear. When selecting genes, the authors should specify the criteria employed to avoid the appearance of cherry-picking. The authors wrote: "We focused on genes differentially expressed with chronological age with a negative Z-score, indicating decreased expression with aging, and mapped these regions where LAD-associated chromatin accessibility was significantly lost ($p < 0.05$) in HGPS compared to healthy controls. A set of genes within A-LADs and B-LADs, that showed reduced chromatin accessibility in HGPS across various immune cell types were pinpointed (Fig. 7a and Supplementary Table 5-6)".

However, I could not find information regarding the dataset and cell types used for the differential gene expression analysis. Additionally, they should report the percentage of overlap between aging-associated dysfunctional genes and the upregulated and downregulated peaks.

10. Figure 6: With a set of interesting genes, it would be beneficial to include their expression profiles, even in a bulk population. RNA-seq data for HGPS patients is available, although not specifically for PBMCs.

Response: We thank the reviewer for suggesting the inclusion of gene expression profiles. However, this study primarily focuses on cell type-specific chromatin accessibility changes in PBMCs from an HGPS patient, assessed through single-cell epigenetic analysis rather than gene expression levels. While RNA-seq data exists for HGPS fibroblasts, it lacks the cell-type specificity necessary for our examination of distinct immune subtypes within PBMCs. For instance, we observed reduced accessibility of AK5, specifically in CD8 Memory cells, whereas genes like ABLIM1 showed consistent changes across various immune cell types. The expression status of AK5 in fibroblasts would not necessarily relate to its regulation in CD8 Memory cells. These findings emphasise the importance of analysing chromatin accessibility at a cell-type-specific level. Given that fibroblast expression data does not capture the complex regulatory landscape of PBMCs, it would not effectively support the immune cell-specific observations of this study.

REVIEWER #2*****

I agree, and for this reason, I believe the paper should refrain from speculating on specific genes altered in HGPS, particularly when highlighting a single gene based on data from a single patient.

11. Figure 6: PRMT6 is located within a Lamin B domain, whereas HGPS results from a mutation in Lamin A. Is the gene situated within a domain shared by both Lamin A and Lamin B? If not, how do the authors comment this detail?

Response: PRMT6 is located exclusively within B-LADs (Lamin B-associated LADs) and not within A-LADs (Lamin A-associated LADs) in the reference datasets we used. Its reduced chromatin accessibility was observed specifically in the promoter regions of CD4 Memory and CD8 Naive T cells. We focused on LAD-associated regions as they represent areas in close contact with the nuclear lamina, making them critical for understanding potential regulatory disruptions linked to nuclear architecture.

Although it remains uncertain how A-LADs and B-LADs are differentially affected in HGPS-derived PBMCs and how these changes influence shifts in chromatin accessibility, our findings suggest that HGPS mutations lead to widespread changes across both A-LAD and B-LAD-associated domains, as well as in predominantly in non-LAD regions (Figure 7c). These observations align with findings by Köhler et al. (Genome Medicine, 2020) in HGPS fibroblasts, where significant chromatin alterations were noted within LADs and largely extended to broader genomic areas. Thus, while PRMT6's reduced accessibility in B-LADs serves as a representative example of specific regulatory changes, the overall nuclear lamina dysfunction in HGPS likely drives complex, genome-wide chromatin remodeling that goes beyond the immediate associations with lamin A/C or lamin B.

REVIEWER #2*****

These observations should also be included in the text to tone down the interpretation of the functional relevance of these findings.

12. Figure 6: The siRNA data is unconvincing to address the role of PRMT6 in premature senescence. As expected, reduced PRMT6 determines cellular senescence as previously described, but its expression is crucial in the HGPS? Why didn't the authors attempt to overexpress PRMT6 in the MSCs differentiated from HGPS patient-derived iPSCs to revert the premature senescence?

Response: We appreciate the reviewer's suggestion to include PRMT6 overexpression experiments. However, the primary aim of this study is to demonstrate the application of IT-scATAC-seq and validate the identification of genes with reduced chromatin accessibility in HGPS-derived PBMCs. We focused on knockdown experiments to investigate the role of PRMT6 in cellular senescence and chromatin state changes, aligning with our methodology and findings.

While we agree that overexpression studies could offer additional insights into reversing the senescence phenotype, such experiments are beyond the current scope, which primarily aims to confirm the downregulation of PRMT6 in the context of stem cell aging. We have also taken care to avoid overinterpreting our results regarding PRMT6's potential to prevent HGPS-associated aging. Future research may explore whether PRMT6 overexpression could rescue HGPS phenotypes in iPSC-derived models, extending the findings reported here.

REVIEWER #2*****

These data should be removed

Reviewer #3

(Remarks to the Author)

Here Wei Jin et al. present IT-scATAC-seq, a new approach for single-cell ATAC-seq. Their method is based on indexed Tn5 tagmentation followed by pooling and distribution into wells for barcoded PCR (a 2-step combinatorial indexing method). They show some benchmarking analysis to compare with other established methods such as 10x, Fluidigm, and sci-ATAC-seq and demonstrate some higher quality metrics. Importantly, the reported cost for IT-scATAC-seq is only around \$0.01/cell, lower than 10x genomics and some other methods. However, this is still higher than the reported cost for EasySciATAC. Overall, the method itself seems to be a sound approach capable of generating high-quality data across a range of sample types.

This work also goes into some detail about the biology of HGPS. To me, this seems to distract from the main message of the paper, the presentation of a novel technology. It is certainly an advantage to show that this technology can work on clinical samples, but the major focus should remain on the evaluation of IT-scATAC-seq as a new technology. More emphasis can be given to the comparison with existing technologies. For example, the authors should discuss limitations of the IT-scATAC technology (the need for indexed Tn5, lower throughput and higher cost relative to sci-ATAC-seq methods (including EasySciATAC), and the need for FANS). Extended data data figure 6 can be moved to the main figure panels to provide the details regarding cost, as lower cost is emphasized as a major advantage of the method. Some of the biological results could be shifted to the supplement. Regarding cost, the cost of Tn5 is currently not included in the calculation.

Overall, this appears to be a sound method for scATAC-seq capable of generating high quality data. Some non-trivial setup is required if labs are wishing to adopt this, most significantly the production of Tn5 and use of FANS, and the cost is not as low as the 3-step combinatorial approaches or the droplet combinatorial indexing approaches (some of which are not mentioned in the paper). In my opinion the in-depth discussion of HGPS only serves to distract from the main message of this paper and should be significantly scaled back in its presentation.

Version 2:

Reviewer comments:

Reviewer #2

(Remarks to the Author)

authors addressed my concerns

Reviewer #3

(Remarks to the Author)

The authors have addressed most of my comments and the revised version presents a much more focused evaluation of the new scATAC-seq technology. I still recommend moving supplementary figure 6 into the main figure panels (this suggestion was not addressed in the response). Supplementary Table 2 serves as a nice summary comparison of the different methods and will be a good resource for labs choosing a methods to suit their needs. However, the presentation of the "Sequencing" column requires some revision. EasySciATAC does not require custom sequencing primers. EasySciATAC and 10x Genomics do not require whole lane sequencing (can multiplex with other libraries).

Reviewer's Comments:

We thank the editor and reviewers for the valuable comments, which allowed us to substantially improve our manuscript. To address the reviewers' comments and strengthen our work, we performed additional mouse embryonic stem cell differentiation experiments to demonstrate the broad application of IT-scATAC-seq. In the revised manuscript, we state that the HGPS PBMC analysis was carried out in only one sample from a progeria patient. Please find detailed point-by-point responses to the reviewers' comments below.

Reviewer #1 (Remarks to the Author)

Jin et al. propose a novel method for scATAC-seq that minimizes cost without compromising quality. Their method, IT-scATAC-seq, utilizes parallel bulk tagmentation to minimize benchtop variation and reduce pipetting and labor time. To test their technique, the authors compared IT-scATAC-seq libraries to libraries from other established scATAC-seq methods. While the sequencing depth was lower in IT-scATAC-seq libraries, they still had high quality, as seen by their high TSS scores, low mitochondrial DNA, and many unique fragments per cell. IT-scATAC-seq could accurately distinguish between different species, cell lines, and cell heterogeneity in human PBMCs.

Using their novel technique, Jin et al. identified cell-type-specific alterations in the epigenetic landscape of HGPS patients when compared to healthy controls. By performing KEGG on loss/gained peaks, they found enrichment in cellular senescence and immune response pathways. Lastly, they identified differentially regulated regions in HGPS PBMCs that correspond with aging, namely PRMT6.

In figure 2b, they mention a higher number of unique fragments per cell in IT-scATAC-seq libraries. It's a bit difficult to see that in the figure, especially compared to the plate method. Are there any sort of statistics/number?

Response: We provide the median number of unique fragments for IT-scATAC-seq and plate-based scATAC-seq here. Based on the matrix from the same cell lines (K562 and HEK293T), we can detect more unique fragments (median) for these cell lines in IT-scATAC-seq (**Response Figure 1**).

In addition, the plate-based method had >95% duplication level in contrast to about 50% duplication level in IT-scATAC-seq. According to a previous report (PMID: 37537502) and PUMATAC (https://github.com/aertslab/PUMATAC_tutorial), higher sequencing depth would give rise to increased library size. Therefore, IT-scATAC-seq would have to give rise to obviously higher unique fragments than plate-based scATAC in the case of a higher sequencing depth.

Response figure 1. Comparison of unique fragments between plate-based and IT methods.

In figure 2d, they show a high FRiP score in IT-scATAC-seq when compared to other methods. Is it possible that the lower sequencing depth is able to artificially inflate the FRiP score?

Since it is a novel method, it seems to be necessary to cross-validate IT-scATAC-seq using CHIP for specific markers and analyze the overlap?

Response: Thank you for raising this important question.

Firstly, we randomly down-sampled our cell lines IT-scATAC-seq pseudo-bulk data to 5%, 25%, 50%, and 75% of reads. Based on the analysis, we found no obvious differences in FRiP between the downsampled libraries (**Response figure 2a**), suggesting that the FRiP score is independent of the size of these IT-scATAC-seq libraries.

To further validate the IT-scATAC-seq signals, we used the H1 cell line and downloaded the H1 ENCODE ChIP-seq datasets (H3K4me3 marks active promoters, H3K27ac marks active enhancers, H3K27me3 marks facultative gene repression, H3K9me3 marks heterochromatin). Strong H3K4me3 and H3K27ac signals are enriched in IT-scATAC-seq, in contrast to a very low signal enrichment for repressive H3K27me3 and H3K9me3 (**Response figure 2b, c**). A whole-chromosome and pluripotency-associated gene track-views also show a good correlation between IT-scATAC and H3K4me3 and H3K27ac, but not H3K27me3 nor H3K9me3 (**Response figure 2d**).

Response figure 2. FRiP score and cross-validation with ChIP-seq data.

They tested their method across species and cell types. Is it possible to perform a time course experiment (in response to differentiation, for example) to determine if IT-scATAC-seq is able to capture changes in chromatin availability across biological events, especially during differentiation or development?

Response: Thank you for your suggestion. To further demonstrate the broad application of IT-scATAC-seq in biological events, we performed a mouse embryonic stem cell differentiation experiment and added the data to the revised manuscript (Main text and Main Figure 3).

I believe the work will be of significance to the field. The authors propose a novel way to perform scATAC-seq that reduces cost and makes the method more accessible to a broader array of labs. Libraries developed from IT-scATAC seq were able to withstand comparison with established scATAC-seq methods in literature.

Response: We appreciate your constructive comments and enthusiastic feedback.

Reviewer #2 (Remarks to the Author):

In this study, authors presented a novel scATAC-seq methodology, the IT-scATAC-seq, ensuring that this new technology provides a scalable, semi-automated workflow capable of preparing libraries for thousands of cells within a single day.

They applied this technique to blood cells from healthy donors, identifying the distinct cell populations based on chromatin accessibility profiles. A comparison between healthy individuals and patients with premature aging revealed significant chromatin accessibility loss across various immune cell types. By integrating this data with lamina-associating domains (LADs), regions of repressive heterochromatin affected in progeria, they identified altered chromatin accessibility in several aging-related genes, including YTHDC1, FXR1, LRRN3, and PRMT6. The decrease in PRMT6 levels was experimentally analysed in another cell model.

Chromatin remodelling is one of the hallmarks of the pathological premature aging and the identification/characterization of molecular mechanisms driving this aberrant process is of interest for the scientific community. Understanding gene regulation at the single-cell level is crucial for studying cellular heterogeneity in complex biological systems, thus the adopted approach is very interesting.

However, a significant limitation of this study is the use of data from a single HGPS patient and two pooled controls, with only one technical replicate. Given the variability in individual epigenetic profiles, this approach is problematic.

I also find the use and comparison of distinct cell sources to be inappropriate, leading to several instances of overinterpretation.

Response: We thank you for the thoughtful assessment and acknowledge the concerns regarding sample limitations. As noted in our manuscript, the primary aim of this study is to showcase IT-scATAC-seq as a novel, scalable approach to single-cell ATAC-seq, demonstrating its potential application with PBMCs from an HGPS patient. The aim of this manuscript is to describe the development of a new technology, and its application on analysing this single HGPS sample serves as a proof of concept, focused on assessing chromatin accessibility patterns and their alignment with known aging-related pathways.

We recognise that using a single HGPS patient sample limits the ability to make broad conclusions about progeroid immunosenescence. HGPS is an exceptionally rare disease, occurring in approximately 1 in every 4 million live births worldwide. The scarcity of cases, along with significant health risks for these patients, makes sample acquisition inherently challenging, and thus, we are not able to obtain samples from more patients at this moment. To emphasise the limitation, the use of only one patient sample is stated in the revised manuscript in the abstract, main text, and especially in the

discussion to clarify that our findings are not generalised to all HGPS cases. To mitigate variability, we have carefully chosen controls and reference datasets, as outlined in our responses to specific comments, ensuring robust comparisons and minimising potential biases in the analysis.

Regarding the comparison of cell sources, we apologise if any unintended ambiguity contributed to the impression of inappropriate use. Our integration of LAD-associated chromatin accessibility follows established methodologies for identifying altered regulatory regions in progeroid cells. We have clarified our rationale in the revised manuscript to prevent misinterpretation.

Additionally, in response to this concern, we have toned down the manuscript title, abstract and relevant description in the results section to avoid implying functional conclusions and to emphasise that our observations are focused on epigenetic changes.

We trust that these clarifications address your main concerns, with further specifics provided in our point-by-point responses below.

Major comments:

1. Figure 1: The differences and advantages of the novel scATAC-seq set up are not clearly explained. The provided scheme and supplementary figures are insufficient. It is also unclear how this method differs from the approach presented by “Cusanovich et al. Cell 2018”.

Response: We thank your feedback and would like to clarify that the differences and advantages of IT-scATAC-seq are detailed throughout the manuscript in multiple sections.

In the **Introduction** part, we discuss the limitations of current scATAC-seq technologies, particularly in terms of the challenges in **costs, sensitivity and scalability**. IT-scATAC-seq was specifically designed to overcome these limitations and disadvantages, providing a **scalable, cost-effective, and adaptable solution** for various research settings.

The **Results** section includes a comparison with conventional scATAC-seq methods in the subsection titled “**Comparing IT-scATAC to other scATAC-seq methods.**” This section highlights the protocol’s advantages in terms of **library complexity, TSS enrichment, and cost**. IT-scATAC-seq enables **high-throughput processing of up to 10,000 cells per day** while minimising reagent costs, achieving up to a 100-fold reduction in per-cell costs compared to traditional plate-based methods. Additionally, **the protocol does not require any equipment specialised for single-cell libraries**(in contrast to 10X and Fluidigm C1) and can be performed in a wide range of laboratories.

In the **Discussion**, we again summarise IT-scATAC-seq's streamlined workflow, which leverages indexed Tn5 bulk tagmentation to reduce labour and handling time. While we do use the Echo 550 liquid handler to further minimise manual pipetting, this instrument is not essential to our protocol. IT-scATAC-seq is fully adaptable and remains efficient and cost-effective even with standard laboratory equipment.

We also summarise its advantages here:

- **Scalability:** IT-scATAC-seq enhances the throughput, increasing cell processing capacity to match high-throughput methods like droplet-based scATAC-seq. The protocol can be scaled by modularly increasing indexed Tn5 transposomes or PCR primers, with minimal oligo synthesis needed, making it suitable for both small- and large-scale applications.
- **Accuracy:** By using cell sorting rather than droplet-based systems, IT-scATAC-seq minimises the doublet rate and avoids barcode misassignment issues commonly associated with droplet and combinatorial indexing methods.
- **Hands-on time:** IT-scATAC-seq requires less manual labour than traditional plate-based scATAC-seq. Using the Echo liquid handler further streamlines pipetting but is not essential to the workflow, allowing labs without access to this equipment still able to implement IT-scATAC-seq efficiently.
- **Cost-effectiveness:** IT scATAC-seq achieves up to a 100-fold reduction in per-cell cost compared to conventional scATAC-seq. The protocol's minimised PCR volume and efficient reagent usage reduce library preparation costs to approximately 0.1 HKD (~\$0.01) per cell, a level of cost-efficiency that is readily achievable with standard lab reagents.

Regarding the difference between the two methods IT-scATAC-seq and sci-ATAC-seq (combinatorial indexing) used in "Cusanovich et al. Cell 2018", we contrast here:

1) Indexing method:

- a. **Single-cell combinatorial Indexing:** This method involves rounds of pooling and splitting with barcoded Tn5 transposases in wells. Nuclei are tagged in bulk in 96-well plates, pooled, and then redistributed into a second set of wells where a second barcode is introduced during PCR. This process relies on the assumption that nuclei indexed in the first round will not be pooled into the same well in the second round, but it still suffers from collisions (doublet) due to the random nature of the pool and splitting.
- b. **IT-scATAC-seq:** We utilize limited parallel labelling for the first-round indexes, followed by flow-cytometry-based single-cell sorting. This ensures that every cell receives a unique barcode, which guarantees high accuracy and low doublet rates. This approach avoids the collisions inherent in combinatorial indexing.

- 2) Scale of Indexed Tn5: sci-ATAC-seq requires the assembly of $96 * N$ indexed Tn5 transposases (e.g., 384 Tn5 complexes when four 96-well plates are used in the first round). Only a few indexed Tn5 transposases are needed for the first round in IT-scATAC-seq (e.g., less than 12), significantly simplifying the process and reducing potential errors. Therefore, the handling of tagmentation in IT-scATAC-seq is much easier than in sci-ATAC-seq.
- 3) Collision: sci-ATAC-seq involves rounds of pooling and splitting with barcoded Tn5 transposases in wells. Nuclei are tagged in bulk in 96-well plates, pooled, and then redistributed into a second set of wells where a second barcode is introduced during PCR. This process relies on the assumption that nuclei indexed in the first round will not be pooled into the same well in the second round, but it still suffers from collisions due to the random nature of pool and splitting. IT-scATAC-seq utilizes limited parallel labelling for the first-round indexes, followed by flow-cytometry-based single-cell sorting. This ensures that every cell receives a unique barcode, which guarantees high accuracy and low doublet rates. This approach avoids the collisions inherent in combinatorial indexing.
- 4) Barcode utilization: > 80% of cell barcodes in sci-ATAC-seq are unused, while all the barcodes in IT-scATAC-seq are used.
- 5) Customized adapters: Considering the Nextera adapter sequences occupy the Tn5 binding sequence, sci-ATAC-seq needs to design customized adapters for sequencing. We integrate the TruSeq adapters into our library to generate a three-round cell indexing strategy in IT-scATAC-seq, which not only increases the throughput flexibility but can also be multiplexed with other samples for the Illumina sequencing platform.

2. *Figure 2: The comparisons presented in the text are overly speculative. The authors stated: "Despite lower sequencing depths (median duplication rate 54-57% for IT-scATAC-seq versus over 95% for plate-based and C1scATAC-seq) (Fig. 2a)", however the duplication rate is not lower than all the presented protocols; or: "Additionally, IT-scATAC-seq produced higher median TSS enrichment scores (Fig. 2e)", but this is true only for the HEK293T whose result is similar to the one obtained by CH-based method.*

Response: We appreciate your feedback and apologise if the presentation of the data was not as clear as intended. We thank the opportunity to clarify our presentation of the data comparing IT-scATAC-seq with other established methods.

Figure 2, panels a and b, compare duplication rates and unique fragments per cell across different scATAC-seq methods, metrics that reflect library complexity and sequencing depth. Generally, higher duplication rates indicate greater sequencing depth, which in turn increases the number of unique fragments up until sequencing depth reaches saturation (Reference: PMID 37537502). By writing,

“Despite lower sequencing depths (median duplication rate 54-57% for IT-scATAC-seq versus over 95% for plate-based and C1 scATAC-seq) (Fig. 2a), IT-scATAC-seq exhibited higher or comparable library complexity, indicated by a higher number of unique fragments per cell (Fig. 2b).”, we described that in our data, although the duplication rates (and thus sequencing depths) for plate-based and C1 methods are much higher than those for IT-scATAC-seq across all cell lines tested, IT-scATAC-seq demonstrates higher (in the case of HEK293T) or similar performance (in the case of K562 and H1) in terms of unique fragments, suggesting that under a lower sequencing depth, we can achieve comparable or higher library complexity.

We have included detailed metrics along with our response comparing the performance of plate-based methods and IT-scATAC-seq as supplementary materials to provide a clear basis for these conclusions.

For the TSS enrichment score, we thank and acknowledge the reviewer’s point and have revised the text to state: “*Additionally, IT-scATAC-seq produced higher or similar median TSS enrichment scores compared with existing methods (Fig. 2e).*”

3. Figure 3: In the IT scATAC-seq analysis of PBMCs shown in Figure 3b, the cell populations are not well-separated, particularly the CD4 subtypes. Does this ATAC-based cell characterization truly reflect genome function? The authors had to rely on scRNA-seq of PBMCs to refine cell identification. Therefore, what is the advantage of using scATAC-seq over scRNA-seq in this context?

Response: Thank you for your question regarding scATAC-seq.

- 1) Reflection of genome function: ATAC-seq is a well-established method for assessing chromatin accessibility, which is a crucial epigenetic aspect of genome function. This technique has been widely used to infer regulatory landscapes behind gene expression dynamically and are not directly observable through transcriptome. The scATAC-seq examples include studies by Buenroostro et al. *Nature*. (2015) (PMID: 26083756), Zhang et al. *Cell*. (2021) (PMID: 34774128), Corces et al. *Nature Genetics*. (2020) (PMID:333106633), and Becker et al. *Nature Genetics* (2022) (PIMD:35726067), where scATAC-seq effectively mapped enhancer and promoter activities across different cell types and conditions like cancer and Alzheimer's.
- 2) Challenges in cell population separation: The separation of closely related cell subtypes, such as CD4 T cell populations, using scATAC-seq can indeed be challenging, a limitation that is well-observed in the scATAC-seq studies. This difficulty primarily stems from the nature of chromatin accessibility data, which may not always provide as distinct markers for cell type differentiation as transcriptomic data. This has been demonstrated by De Rop et al. (2024, PMID: 37537502), in their systematic benchmarking of scATAC-seq technologies using human PBMCs; similarly when Mezger et al. (2018, PMID: 30194434) developed a scATAC-seq technology and applied it to

human PBMCs, as well as many studies analyzing aging, aging-related conditions, and infections such as COVID-19 using human PBMCs have also demonstrated these limitations (e.g., PMID: 32780218, 37081034, PMID: 37095391). In this study, the annotation between naive and memory CD4 T cells was based on integrated scRNA-seq data and specific cell markers.

- 3) Integration with scRNA-seq: We did not solely rely on scRNA-seq for cell population annotation. Integrating scRNA-seq data is one common approach to enhance the annotation of cell types identified by scATAC-seq, utilizing gene expression profiles to corroborate and refine chromatin accessibility-based predictions (PMID:34725479,33106633). Manual annotation based on known marker genes is also viable and is employed where appropriate, but integration with scRNA-seq can provide a more robust and detailed cell type classification, particularly for closely related subtypes. We also showed the enhanced accessibility of cell type-specific markers of different populations in **Figure 3c**.
- 4) Advantages of scATAC-seq over scRNA-seq: While scRNA-seq is a powerful method for capturing gene expression profiles, scATAC-seq provides unique and complementary information by directly measuring chromatin accessibility, thereby revealing the regulatory landscape that drives gene expression changes. Previous works have demonstrated the epigenetic deregulation in HGPS cells, but no epigenetic results have ever been reported for patient's blood cells. In the context of our study, scATAC-seq allows us to pinpoint active regulatory elements, such as enhancers and promoters, and to identify specific chromatin accessibility patterns associated with HGPS that influence gene regulation beyond the transcriptional layer. For instance, we observed LAD-associated changes in chromatin accessibility—an indication of nuclear lamina disruptions common in progeroid syndromes like HGPS—that are inherently epigenetic and not detectable by transcriptional profiling alone. This does not imply that scRNA-seq is less valuable in this context; indeed, it remains a powerful tool for disease research. However, the focus of our current work is on demonstrating the broad utility of our new scATAC-seq technology and providing novel insights into the *epigenetic* alterations associated with HGPS, which is beyond what scRNA-seq can reveal. We agree that although exploring the expression patterns in HGPS is beyond the scope of this study, such analysis would be a valuable direction for future research.

4. Figure 3: The authors compared their IT-scATAC-seq with two available PBMC scATAC-seq datasets generated using different methods (WT 10X and WT s3). However, the rationale for selecting only these two controls is unclear, especially given the eight different scATAC-seq protocols mentioned in De Rop et al., Nat Biotech 2023, which the authors cite.

Response: Thank you for your question regarding our selection of control datasets in Figure 3, and I apologise for not being clearer in the explanation of the rationale. In our comparison, we opted to use datasets from the study by De Rop et al., Nat Biotech 2023, which extensively characterised PBMCs using eight different scATAC-seq methods: 10x Genomics scATAC-seq (v1, v1.1, v2, multiome and mitochondrial scATAC (mtscATAC)) as well as Bio-Rad ddSEQ, HyDrop and s3-ATAC. The same PBMC cell source (pooled two healthy samples) was used across eight different methods:

“Cryopreserved human PBMCs from one male donor and one female donor were purchased from AllCells and distributed across institutes to generate the following samples.....”.

The eight methods in the report by De Rop et al. are primarily categorised into two groups: droplet-based (10X v1, v1.1, v2, ddSEQ, HyDrop) and sci-based protocols (s3) (10X multiome and mtscATAC are multi-omics profiling, and mitochondrial profiling is a variant of 10X that profiles other modalities rather than chromatin accessibility). We selected one representative dataset from each category—10X v1.1 for droplet-based and s3 for sci-based—to provide a broad perspective on the most prevalent technologies in the field. By incorporating these datasets, we aimed to minimise discrepancies arising from both distinct technological approaches and potential biological variability. In other words, using one dataset from our own research as a biological WT reference, alongside the two additional WT datasets from De Rop et al. processed by different technologies, ensures our comparison between HGPS and WT PBMCs is robust, minimising the impact of both technological and biological variations.

5. Figure 4: The interpretation of the differences between the controls and the single HGPS sample is challenging to follow. Initially, the authors mention an overall decrease in chromatin accessibility but then shift focus to upregulated peaks. However, several downregulated peaks are also linked to T lymphocyte activation, such as the mTOR-signalling pathway. How do the authors justify this discrepancy? They wrote: *“Functional analysis of down-regulated peak-associated genes revealed enrichment in immune response pathways, including cell adhesion, lymphocyte differentiation, chemokine, signalling, cytokine production, and infection-related pathways”.* However, cytokine production, signaling, lymphocyte differentiation are terms characterizing the immune system activation. Notably, one of the KEGG terms associated with the downregulated peaks is *“cellular senescence”.* They concluded: *“cellular senescence pathways were significantly enriched across all cell*

types aligning with accelerated immunosenescence in HGPS observed in previous studies”, however, this conclusion appears to be an overinterpretation based on the GO analysis in Figure 4b.

Response: We thank you for your detailed comments and would like to clarify several aspects regarding our analysis and interpretation of chromatin accessibility changes in HGPS PBMCs.

Our analysis identified the observed overall decrease in chromatin accessibility across all immune cell populations analysed in HGPS (**Figure 5a**). Contrary to “shift focus to upregulated peaks”, our study examines both regions with increased accessibility (up-regulated peaks) and regions with decreased accessibility (down-regulated peaks) (**Figure 5b-c, Extended Data Fig.8**), presenting a balanced view of chromatin state alterations in HGPS PBMCs.

For regions with decreased chromatin accessibility (*down-regulated peaks*, **Figure 5c**), we conducted an enrichment analysis to identify the pathways and processes potentially affected by *LMNA* mutation in HGPS patient’s PBMCs. Genes associated with these downregulated peaks (2.5kb ± of TSS region) were found to be enriched in pathways related to cell adhesion, lymphocyte differentiation, chemokine signalling, cytokine production, and infection responses. This does not indicate or imply immune activation in these populations. Rather, it suggests that genes (could be pro- or anti-cell adhesion, lymphocyte differentiation, infection, cytokine production) that are involved in these immune-related processes exhibited reduced accessibility, potentially *affecting* the regulation of these pathways. Notably, previous studies on lamin A/C-related immune cell function support this interpretation, showing that lamin A/C mutations or deficiencies impact immune regulation, leading to *altered* immune signaling, activation, and aging-related immune deficits, which has been reviewed by Saez et al. (2020, PMID: 32854281).

Regarding the KEGG “Cellular Senescence” pathway (hsa04218), we observed that this pathway was enriched among *down-regulated* peaks across all analyzed cell populations. It is important to note that hsa04218 includes both pro-senescent and anti-senescent genes (<https://www.genome.jp/pathway/hsa04218>). Thus, our analysis only shows that the accessibility (expression regulation) of cellular senescence-related genes is reduced in HGPS PBMCs, which are implicated in HGPS-linked aging phenotypes. This finding does not imply that “cellular senescence” itself is downregulated in these cells, nor does it suggest a causal relationship between decreased accessibility of these genes and senescence phenotypes. We believe this provides valuable insight into the chromatin state changes associated with accelerated aging in HGPS without overinterpreting the data.

Together, our findings reveal a pattern of chromatin accessibility changes in HGPS PBMCs that aligns with documented immunosenescence features, including reduced accessibility in genes linked to

immune response pathways and cellular senescence. To avoid inferring any functional implications of these analyses, we have also revised the description of the relevant parts in the manuscript to emphasise the findings that were made for the epigenetic changes. We hope this clarifies the confusion and addresses your concerns about the data interpretation.

6. Figure 4: Are the upregulated ATAC-seq peaks in HGPS-derived PBMC indicative of a specific epigenetic chromatin state, or are they also linked to active transcription? Expression analysis is essential for any interpretation of function and dysfunction.

Response: We appreciate your interest in the relationship between ATAC-seq peaks and active transcription.

Upregulated ATAC-seq peaks reflect regions of increased chromatin accessibility at regulatory elements such as TSS, enhancers, and promoters, which suggest the potential for transcriptional regulation but do not directly measure gene expression.

Importantly, ATAC-seq and scRNA-seq capture distinct regulatory and expression-level information, each providing unique insights into cellular states. In this study, the primary aim is to use IT-scATAC-seq to explore epigenetic chromatin states in HGPS-derived PBMCs, focusing specifically on chromatin accessibility as an indicator of regulatory potential rather than gene expression.

To gain a complete understanding of cellular function and dysfunction (which we neither covered nor intended to cover in this study), expression analysis alone is often insufficient. scRNA-seq, for example, is limited by its reliance on polyadenylated RNA capture, which restricts transcriptomic coverage and omits certain RNA species. Furthermore, gene expression levels do not reliably predict protein abundance, as post-transcriptional modifications and other regulatory mechanisms play crucial roles in shaping cellular functions.

We recognise the value of a multimodal approach to achieve a fuller understanding of chromatin state and transcriptional activity and plan to explore this in future studies as techniques evolve further.

7. Figure 5: The authors chose to intersect the differentially accessible peaks with LAD mapped in HeLa and HAP-1 cells. While most of the LADs are conserved, they opted for a cancer cell line, which likely has an altered chromatin structure, while HAP-1 is a haploid cell line. Why didn't they consider using LADs from a human primary cell line, such as fibroblasts? In the paper cited by the authors, Kohler et al. (Genome Med. 2020) present LAD data for both wild-type and HGPS fibroblasts. Why did the authors not consider the HGPS LADs instead of those from tumor cells? Additionally, Kohler et al. also provide ATAC-seq data; how are the pathological euchromatin changes conserved between fibroblasts and PBMCs? Are they conserved or cell specific?

Response: We thank you for your question regarding LAD dataset selection.

LAD data from primary fibroblasts are indeed scarce due to the limited availability of high-quality lamin A/C ChIP-seq data from these cells. Yet, LAD regions are known to be relatively conserved across cell types Kind et al. Cell. 2015 (PMID: 26365489). It is worth noting that in the cited study by Kohler et al. (Genome Med. 2020), they did not present LAD data for wildtype nor HGPS fibroblast; instead, they also relied on LAD data from HeLa cells rather than directly profiling LADs in fibroblasts, due to the high conservation of LAD domains across cell types.

“For the comparison of LAD- and “solo-WCGW” CpG probe methylation levels between HGPS and control samples, we used previously published locations of lamin A LADs (PMID: 25602132), lamin B LADs (PMID: 18463634) ...”

Following what has been used by Kohler et al., we used the same HeLa-derived LAD dataset for A-LADs. For B-LADs, we used the dataset generated from K562 cells (PMID: 32893442).

In our analysis, we observed cell-type-specific changes in chromatin accessibility across different immune cell types within HGPS-derived PBMCs, revealing euchromatin alteration signatures unique to each subtype. As for comparison with fibroblast data, Köhler et al. reported relatively fewer differentially accessible peaks in HGPS fibroblasts than we have identified in PBMCs. It is difficult to determine whether these differences are due to cell type or variations in analytical methods.

Interestingly, FOS2 and BACH2 (among other AP1 family TFs), enriched in differentially accessible regions in HGPS fibroblasts identified in Köhler et al. study, were also found enriched in CD4 Memory cells of in the HGPS patient (Supplementary Figure 9b).

8. Figure 5: the authors stated Page 10 line 274: “Notably, most immune cell types showed a marked loss of peaks in both A-LADs and B-LADs, but CD4 naive T cells exhibited more gained A-LADs-associated peaks (393 gained vs. 297 lost) and comparable alterations in B-LADs-associated peaks (1,146 gained vs. 1,194 lost), suggesting cell-type-specific resilience to HGPS-induced chromatin alterations”, however, this is an overinterpretation given the lack of Lamin ChIP-seq data in these cells.

Response: We thank you for your comments and acknowledge that Lamin A/C ChIP-seq data would provide a more direct assessment of LAD structural changes in HGPS. However, our study’s objective is to investigate chromatin accessibility changes in specific cell types in HGPS PBMCs, with LAD regions (which are conserved among human cell types) of particular interest, rather than to examine broader structural changes in LAD domains themselves.

Our statement regarding “resilience” in CD4 naive T cells reflects the observation that, unlike other immune cell types, these cells exhibited a unique pattern of peak gains in A-LADs (393 gained vs. 297 lost). This pattern suggests a distinct chromatin accessibility change pattern within the LADs region of CD4 naive T cells under HGPS conditions rather than implying structural resilience in LAD regions. We have tone-downed the text to clarify that this interpretation is based on observed chromatin accessibility patterns within LAD-associated regions relative to other types of cells and does not imply direct structural changes in LADs of patient’s cell populations without ChIP-seq confirmation.

9. Figure 6: How did the authors select the set of genes shown in Figure 6a? Are all the genes with reduced chromatin accessibility in LAD across various cell populations included, or did they choose only the most interesting ones?

Response: We thank you for the question regarding the selection criteria for the set of genes shown in Figure 6a. The genes depicted are representative of regions with altered chromatin accessibility in LADs across various cell populations, specifically those that overlap with findings from a previously published aging-related study. As outlined in the manuscript, we focused on genes that exhibited significant loss of chromatin accessibility in LADs ($p < 0.05$) and correlated with decreased expression associated with chronological aging. This targeted approach ensured that we included genes with known relevance to both HGPS and aging.

To clarify, the complete list of overlapped genes with significantly altered accessibility is detailed in **Supplementary Data 5** and **6**. Among them, we selected a subset that best represented the key findings of our study.

The above has also been noted in the original manuscript: *“To identify regions in HGPS PBMCs that are differentially regulated and associated with aging, we compared chromatin accessibility changes in HGPS samples with genes known to be affected by chronological aging. We focused on genes differentially expressed with chronological age with a negative Z-score, indicating decreased expression with aging, and mapped these regions where LAD-associated chromatin accessibility was significantly lost in HGPS compared to healthy controls”*.

10. Figure 6: With a set of interesting genes, it would be beneficial to include their expression profiles, even in a bulk population. RNA-seq data for HGPS patients is available, although not specifically for PBMCs.

Response: We thank the reviewer for suggesting the inclusion of gene expression profiles. However, this study primarily focuses on cell type-specific chromatin accessibility changes in PBMCs from an HGPS patient, assessed through single-cell epigenetic analysis rather than gene expression levels. While RNA-seq data exists for HGPS fibroblasts, it lacks the cell-type specificity necessary for our examination of distinct immune subtypes within PBMCs. For instance, we observed reduced accessibility of AK5, specifically in CD8 Memory cells, whereas genes like ABLIM1 showed consistent changes across various immune cell types. The expression status of AK5 in fibroblasts would not necessarily relate to its regulation in CD8 Memory cells. These findings emphasise the importance of analysing chromatin accessibility at a cell-type-specific level. Given that fibroblast expression data does not capture the complex regulatory landscape of PBMCs, it would not effectively support the immune cell-specific observations of this study.

11. Figure 6: PRMT6 is located within a Lamin B domain, whereas HGPS results from a mutation in Lamin A. Is the gene situated within a domain shared by both Lamin A and Lamin B? If not, how do the authors comment this detail?

Response: PRMT6 is located exclusively within B-LADs (Lamin B-associated LADs) and not within A-LADs (Lamin A-associated LADs) in the reference datasets we used. Its reduced chromatin accessibility was observed specifically in the promoter regions of CD4 Memory and CD8 Naive T cells. We focused on LAD-associated regions as they represent areas in close contact with the nuclear lamina, making them critical for understanding potential regulatory disruptions linked to nuclear architecture.

Although it remains uncertain how A-LADs and B-LADs are differentially affected in HGPS-derived PBMCs and how these changes influence shifts in chromatin accessibility, our findings suggest that

HGPS mutations lead to widespread changes across both A-LAD and B-LAD-associated domains, as well as in predominantly in non-LAD regions (**Figure 7c**). These observations align with findings by Köhler et al. (Genome Medicine, 2020) in HGPS fibroblasts, where significant chromatin alterations were noted within LADs and largely extended to broader genomic areas. Thus, while PRMT6's reduced accessibility in B-LADs serves as a representative example of specific regulatory changes, the overall nuclear lamina dysfunction in HGPS likely drives complex, genome-wide chromatin remodeling that goes beyond the immediate associations with lamin A/C or lamin B.

12. Figure 6: The siRNA data is unconvincing to address the role of PRMT6 in premature senescence. As expected, reduced PRMT6 determines cellular senescence as previously described, but its expression is crucial in the HGPS? Why didn't the authors attempt to overexpress PRMT6 in the MSCs differentiated from HGPS patient-derived iPSCs to revert the premature senescence?

Response: We appreciate the reviewer's suggestion to include PRMT6 overexpression experiments. However, the primary aim of this study is to demonstrate the application of IT-scATAC-seq and validate the identification of genes with reduced chromatin accessibility in HGPS-derived PBMCs. We focused on knockdown experiments to investigate the role of PRMT6 in cellular senescence and chromatin state changes, aligning with our methodology and findings.

While we agree that overexpression studies could offer additional insights into reversing the senescence phenotype, such experiments are beyond the current scope, which primarily aims to confirm the downregulation of PRMT6 in the context of stem cell aging. We have also taken care to avoid overinterpreting our results regarding PRMT6's potential to prevent HGPS-associated aging. Future research may explore whether PRMT6 overexpression could rescue HGPS phenotypes in iPSC-derived models, extending the findings reported here.

Minor comments:

1) *In several panels, the text is too small to be readable on a printed page. Examples: Figure 1J, Extended Figure 5a, Figure 4b, figure 4c.*

Response: Thank you for this suggestion. We have revised the front size accordingly.

2) *Page 6 line 129 "...20 randomly selected single-cell profiles demonstrated high congruence with bulk data, with Pearson correlation coefficients ranging from 0.71 to 0.94". However, the Extended Data Fig. 5a shows a correlation ranging from 0.52 to 0.94*

Response: Thank you for pointing out this error and we have corrected the manuscript accordingly.

3) *The order of protocols presented on y axis of distinct panels of Figure 2 is not the same for all, making it challenging to follow.*

Response: Thank you for the suggestion. We have adapted the display orders to make it more readable and consistent.

4) *One of the claims is that the protocol can be completed in a single day; however, no information about the other protocols was provided.*

Response: We appreciate the reviewer's observation. Our primary objective in stating that the protocol can process 10,000 cells in a single day is to emphasize the efficiency and scalability of IT-scATAC-seq. We did not intend to compare the total timeline with other protocols in this study, as such comparisons fall outside the scope of our work. Rather, our statement illustrates IT-scATAC-seq's capacity for high-throughput single-cell chromatin profiling, achievable with a streamlined setup.

5) *Another assertion made by the authors is the simplicity of the protocol and the absence of overly complicated or expensive instruments. However, the automated Labcyte Echo 55 Acoustic liquid handler is quite costly and is not available in every laboratory.*

Response: We acknowledge that the use of the Echo 550 Acoustic Liquid Handler accelerates our workflow by minimizing manual handling, but it is not essential to our IT-scATAC-seq protocol. Our strategy remains fully adaptable and cost-effective without the Echo 550, providing a robust framework for single-cell omics that can be implemented in a wide range of laboratory settings with standard equipment.

Reviewer's Comments:

We sincerely appreciate the time and effort the reviewers have dedicated to evaluating our manuscript, providing constructive feedback, and providing us with the opportunity to improve our work.

We carefully considered all the suggestions from you and the editors and revised the manuscript to improve its focus, clarity, and scientific rigour. The manuscript is now restructured to highlight IT-scATAC-seq as a novel, scalable, and cost-effective approach for single-cell chromatin accessibility profiling, with benchmarking and validation against existing methods.

Below, we provide a point-by-point response in **blue** and previous responses in **grey**.

REVIEWER COMMENTS

Reviewer #1 (Remarks to the Author):

The authors have addressed my concerns.

Response: We appreciate your valuable feedback and are glad to hear that your concerns have been addressed. Thank you for your time and constructive comments.

Reviewer #2 (Remarks to the Author):

In this study, authors presented a novel scATAC-seq methodology, the IT-scATAC-seq, ensuring that this new technology provides a scalable, semi-automated workflow capable of preparing libraries for thousands of cells within a single day. They applied this technique to blood cells from healthy donors, identifying the distinct cell populations based on chromatin accessibility profiles. A comparison between healthy individuals and patients with premature aging revealed significant chromatin accessibility loss across various immune cell types. By integrating this data with lamina-associating domains (LADs), regions of repressive heterochromatin affected in progeria, they identified altered chromatin accessibility in several aging-related genes, including YTHDC1, FXR1, LRRN3, and PRMT6. The decrease in PRMT6 levels was experimentally analysed in another cell model. Chromatin remodelling is one of the hallmarks of the pathological premature aging and the identification/characterization of molecular mechanisms driving this aberrant process is of interest for the scientific community. Understanding gene regulation at the single-cell level is crucial for studying cellular heterogeneity in complex biological systems, thus the adopted approach is very interesting. However, a significant limitation of this study is the use of data from a single HGPS patient and two pooled controls, with only one technical replicate. Given the variability in individual epigenetic profiles, this approach is problematic. I also find the use and comparison of distinct cell sources to be inappropriate, leading to several instances of overinterpretation.

1st round of Response: We thank you for the thoughtful assessment and acknowledge the concerns regarding sample limitations. As noted in our manuscript, the primary aim of this study is to showcase IT-scATAC-seq as a novel, scalable approach to single-cell ATAC-seq, demonstrating its potential application with PBMCs from an HGPS patient. The aim of this manuscript is to describe the development of a new technology, and its application on analysing this single HGPS sample serves as a proof of concept, focused on assessing chromatin accessibility patterns and their alignment with known aging-related pathways. We recognise that using a single HGPS patient sample limits the ability to make broad conclusions about progeroid immunosenescence. HGPS is an exceptionally rare disease, occurring in approximately 1 in every 4 million live births worldwide. The scarcity of cases, along with significant health risks for these patients, makes sample acquisition inherently challenging, and thus, we are not able to obtain samples from more patients at this moment.

To emphasise the limitation, the use of only one patient sample is stated in the revised manuscript in the abstract, main text, and especially in the discussion to clarify that our findings are not generalised to all HGPS cases. To mitigate variability, we have carefully chosen controls and reference datasets, as outlined in our responses to specific comments, ensuring robust comparisons and minimising potential biases in the analysis.

Regarding the comparison of cell sources, we apologise if any unintended ambiguity contributed to the impression of inappropriate use. Our integration of LAD-associated chromatin accessibility follows established methodologies for identifying altered regulatory regions in progeroid cells. We have clarified our rationale in the revised manuscript to prevent misinterpretation.

Additionally, in response to this concern, we have toned down the manuscript title, abstract and relevant description in the results section to avoid implying functional conclusions and to emphasise that our observations are focused on epigenetic changes.

We trust that these clarifications address your main concerns, with further specifics provided in our point-by-point responses below.

REVIEWER #2*****

Overall, I find this revision insufficient, as none of my concerns have been adequately addressed. The responses provided are more structured but largely reiterate arguments already present in the original text. Below, a detailed point-by-point commentary.

Number of HGPS patients:

In response to my request to include more HGPS patients, the authors stated that the rarity of the disease limits material availability. They wrote in their rebuttal: “The scarcity of cases, along with significant health risks for these patients, makes sample acquisition inherently challenging, and thus, we are not able to obtain samples from more patients at this moment” However, the Progeria Research Foundation (<https://www.progeriaresearch.org/>) offers several cell lines for scientific purposes. These include multiple lymphoblast cell lines which, although not identical to PBMCs, could be used for B cell validation experiments.

Comparison with cancer cells:

This point remains unaddressed. Despite the availability of several datasets on LADs in healthy cells, the authors did not use other LAD datasets. They wrote: “Our integration of LAD-associated chromatin accessibility follows established methodologies for identifying

altered regulatory regions in progeroid cells” However, this response does not address my concern, which was about the rationale behind the comparison, not the methodology used in the analysis.

The progeria focus:

The central theme of the paper remains predominantly centered on progeria, with four out of seven figures dedicated to this topic. In my view, the data provided does not sufficiently support this focus. If the intent of this work is to promote the technology, I would be open to reconsider my position. However, the sections addressing the HGPS sample should be presented as descriptive, with explicit explanations provided in the text.

Response: We deeply appreciate your thoughtful review and the time you have dedicated to evaluating our manuscript. We acknowledge the concern regarding the limited HGPS sample size and its implications for drawing biological conclusions. As this work is fundamentally a methodological innovation study, our primary aim is to establish the technical robustness, accessibility, and scalability of IT-scATAC-seq rather than to address specific biological hypotheses. In alignment with this goal, we have removed all HGPS-related data and discussions to refocus the manuscript on methodological validation and utility for the community. To accelerate the utility and accessibility of this method to more labs, we also enclosed a detailed step-by-step protocol in the **Supplementary Method**.

Thank you again for your insightful feedback, which has strengthened the clarity and purpose of our work.

1. Figure 1: The differences and advantages of the novel scATAC-seq set up are not clearly explained. The provided scheme and supplementary figures are insufficient. It is also unclear how this method differs from the approach presented by “Cusanovich et al. Cell 2018”.

1st round of response: We thank your feedback and would like to clarify that the differences and advantages of IT-scATAC-seq are detailed throughout the manuscript in multiple sections.

In the Introduction part, we discuss the limitations of current scATAC-seq technologies, particularly in terms of the challenges in costs, sensitivity and scalability. IT-scATAC-seq was specifically designed to overcome these limitations and disadvantages, providing a scalable, cost-effective, and adaptable solution for various research settings.

The Results section includes a comparison with conventional scATAC-seq methods in the subsection titled “Comparing IT-scATAC to other scATAC-seq methods.” This section highlights the protocol’s advantages in terms of library complexity, TSS enrichment, and cost. IT-scATAC-seq enables high-throughput processing of up to 10,000 cells per day while minimising reagent costs, achieving up to a 100-fold reduction in per-cell costs compared to traditional plate-based methods. Additionally, the protocol does not require any equipment specialised for single-cell libraries (in contrast to 10X and Fluidigm C1) and can be performed in a wide range of laboratories.

In the Discussion, we again summaries IT-scATAC-seq’s streamlined workflow, which leverages indexed Tn5 bulk tagmentation to reduce labor and handling time. While we do use the Echo 550 liquid handler to further minimize manual pipetting, this instrument is not essential to our protocol. IT-scATAC-seq is fully adaptable and remains efficient and cost-effective even with standard laboratory equipment.

We also summaries its advantages here:

Scalability: IT-scATAC-seq enhances the throughput, increasing cell processing capacity to match high-throughput methods like droplet-based scATAC-seq. The protocol can be scaled by modularly increasing indexed Tn5 transposomes or PCR primers, with minimal oligo synthesis needed, making it suitable for both small- and large-scale applications.

Accuracy: By using cell sorting rather than droplet-based systems, IT-scATAC-seq minimises the doublet rate and avoids barcode misassignment issues commonly associated with droplet and combinatorial indexing methods.

Hands-on time: IT-scATAC-seq requires less manual labour than traditional plate-based scATAC-seq. Using the Echo liquid handler further streamlines pipetting but is not essential to the workflow, allowing labs without access to this equipment still able to implement IT-scATAC-seq efficiently.

Cost-effectiveness: IT scATAC-seq achieves up to a 100-fold reduction in per-cell cost compared to conventional scATAC-seq. The protocol’s minimised PCR volume and efficient reagent usage reduce library preparation costs to approximately 0.1 HKD (~\$0.01) per cell, a level of cost-efficiency that is readily achievable with standard lab reagents.

Regarding the difference between the two methods IT-scATAC-seq and sci-ATAC-seq (combinatorial indexing) used in “Cusanovich et al. Cell 2018”, we contrast here:

Indexing method:

Single-cell combinatorial Indexing: This method involves rounds of pooling and splitting with barcoded Tn5 transposases in wells. Nuclei are tagged in bulk in 96-well plates, pooled,

and then redistributed into a second set of wells where a second barcode is introduced during PCR. This process relies on the assumption that nuclei indexed in the first round will not be pooled into the same well in the second round, but it still suffers from collisions (doublet) due to the random nature of the pool and splitting.

IT-scATAC-seq: We utilize limited parallel labelling for the first-round indexes, followed by flow-cytometry-based single-cell sorting. This ensures that every cell receives a unique barcode, which guarantees high accuracy and low doublet rates. This approach avoids the collisions inherent in combinatorial indexing.

Scale of Indexed Tn5: sci-ATAC-seq requires the assembly of $96 * N$ indexed Tn5 transposases (e.g., 384 Tn5 complexes when four 96-well plates are used in the first round). Only a few indexed Tn5 transposases are needed for the first round in IT-scATAC-seq (e.g., less than 12), significantly simplifying the process and reducing potential errors. Therefore, the handling of tagmentation in IT-scATAC-seq is much easier than in sci-ATAC-seq.

Collision: sci-ATAC-seq involves rounds of pooling and splitting with barcoded Tn5 transposases in wells. Nuclei are tagged in bulk in 96-well plates, pooled, and then redistributed into a second set of wells where a second barcode is introduced during PCR. This process relies on the assumption that nuclei indexed in the first round will not be pooled into the same well in the second round, but it still suffers from collisions due to the random nature of pool and splitting. IT-scATAC-seq utilizes limited parallel labelling for the first-round indexes, followed by flow-cytometry-based single-cell sorting. This ensures that every cell receives a unique barcode, which guarantees high accuracy and low doublet rates. This approach avoids the collisions inherent in combinatorial indexing.

Barcode utilization: > 80% of cell barcodes in sci-ATAC-seq are unused, while all the barcodes in IT-scATAC-seq are used.

Customized adapters: Considering the Nextera adapter sequences occupy the Tn5 binding sequence, sci-ATAC-seq needs to design customized adapters for sequencing. We integrate the TruSeq adapters into our library to generate a three-round cell indexing strategy in IT-scATAC-seq, which not only increases the throughput flexibility but can also be multiplexed with other samples for the Illumina sequencing platform.

REVIEWER #2*****

The advantages of the technology outlined by the authors in the first part of their response were already evident from the original text. Differences from the methodology described in

Cusanovich et al., Cell (2018) are now clearer; it would be helpful to summarize these distinctions in a supplementary table for the benefit of readers who are not experts in ATAC-seq.

Response: We thank you for this suggestion. We now include a contrast with previous methods, including the sci-based method (Cusanovich et al., Cell (2018)), together in a table in **Supplementary Data 2** for clearer presentation.

2. Figure 2: The comparisons presented in the text are overly speculative. The authors stated: “Despite lower sequencing depths (median duplication rate 54-57% for IT-scATAC-seq versus over 95% for plate-based and C1scATAC-seq) (Fig. 2a)”, however the duplication rate is not lower than all the presented protocols; or: “Additionally, IT-scATAC-seq produced higher median TSS enrichment scores (Fig. 2e)”, but this is true only for the HEK293T whose result is similar to the one obtained by CH-based method.

1st round of response: We appreciate your feedback and apologise if the presentation of the data was not as clear as intended. We thank the opportunity to clarify our presentation of the data comparing IT-scATAC-seq with other established methods.

Figure 2, panels a and b, compare duplication rates and unique fragments per cell across different scATAC-seq methods, metrics that reflect library complexity and sequencing depth. Generally, higher duplication rates indicate greater sequencing depth, which in turn increases the number of unique fragments up until sequencing depth reaches saturation (Reference: PMID 37537502).

By writing, “Despite lower sequencing depths (median duplication rate 54-57% for IT-scATAC-seq versus over 95% for plate-based and C1 scATAC-seq) (Fig. 2a), IT-scATAC-seq exhibited higher or comparable library complexity, indicated by a higher number of unique fragments per cell (Fig. 2b).”, we described that in our data, although the duplication rates (and thus sequencing depths) for plate-based and C1 methods are much higher than those for IT-scATAC-use across all cell lines tested, IT-scATAC-seq demonstrates higher (in the case of HEK293T) or similar performance (in the case of K562 and H1) in terms of unique fragments, suggesting that under a lower sequencing depth, we can achieve comparable or higher library complexity.

We have included detailed metrics along with our response comparing the performance of plate-based methods and IT-scATAC-seq as supplementary materials to provide a clear basis for these conclusions.

For the TSS enrichment score, we thank and acknowledge the reviewer's point and have revised the text to state: "Additionally, IT-scATAC-seq produced higher or similar median TSS enrichment scores compared with existing methods (Fig. 2e)."

REVIEWER #2*****

I appreciated the revision of the text

Response: Thanks for this comment.

3. Figure 3: In the IT scATAC-seq analysis of PBMCs shown in Figure 3b, the cell populations are not well-separated, particularly the CD4 subtypes. Does this ATAC-based cell characterization truly reflect genome function? The authors had to rely on scRNA-seq of PBMCs to refine cell identification. Therefore, what is the advantage of using scATAC-seq over scRNA-seq in this context?

1st round of response: Thank you for your question regarding scATAC-seq.

1) Reflection of genome function: ATAC-seq is a well-established method for assessing chromatin accessibility, which is a crucial epigenetic aspect of genome function. This technique has been widely used to infer regulatory landscapes behind gene expression dynamically and are not directly observable through transcriptome. The scATAC-seq examples include studies by Buenrostro et al. *Nature*. (2015) (PMID: 26083756), Zhang et al. *Cell*. (2021) (PMID: 34774128), Corces et al. *Nature Genetics*. (2020) (PMID:333106633), and Becker et al. *Nature Genetics* (2022) (PIMD:35726067), where scATAC-seq effectively mapped enhancer and promoter activities across different cell types and conditions like cancer and Alzheimer's.

2) Challenges in cell population separation: The separation of closely related cell subtypes, such as CD4 T cell populations, using scATAC-seq can indeed be challenging, a limitation that is well-observed in the scATAC-seq studies. This difficulty primarily stems from the nature of chromatin accessibility data, which may not always provide as distinct markers for cell type differentiation as transcriptomic data. This has been demonstrated by De Rop et al.

(2024, PMID: 37537502), in their systematic benchmarking of scATAC-seq technologies using human PBMCs; similarly when Mezger et al. (2018, PMID: 30194434) developed a scATAC-seq technology and applied it to human PBMCs, as well as many studies analyzing aging, aging-related conditions, and infections such as COVID-19 using human PBMCs have also demonstrated these limitations (e.g., PMID: 32780218, 37081034, PMID: 37095391). In this study, the annotation between naive and memory CD4 T cells was based on integrated scRNA-seq data and specific cell markers.

3) Integration with scRNA-seq: We did not solely rely on scRNA-seq for cell population annotation. Integrating scRNA-seq data is one common approach to enhance the annotation of cell types identified by scATAC-seq, utilizing gene expression profiles to corroborate and refine chromatin accessibility-based predictions (PMID:34725479,33106633). Manual annotation based on known marker genes is also viable and is employed where appropriate, but integration with scRNA-seq can provide a more robust and detailed cell type classification, particularly for closely related subtypes. We also showed the enhanced accessibility of cell type-specific markers of different populations in Figure 3c.

4) Advantages of scATAC-seq over scRNA-seq: While scRNA-seq is a powerful method for capturing gene expression profiles, scATAC-seq provides unique and complementary information by directly measuring chromatin accessibility, thereby revealing the regulatory landscape that drives gene expression changes. Previous works have demonstrated the epigenetic deregulation in HGPS cells, but no epigenetic results have ever been reported for patient's blood cells. In the context of our study, scATAC-seq allows us to pinpoint active regulatory elements, such as enhancers and promoters, and to identify specific chromatin accessibility patterns associated with HGPS that influence gene regulation beyond the transcriptional layer. For instance, we observed LAD-associated changes in chromatin accessibility—an indication of nuclear lamina disruptions common in progeroid syndromes like HGPS—that are inherently epigenetic and not detectable by transcriptional profiling alone. This does not imply that scRNA-seq is less valuable in this context; indeed, it remains a powerful tool for disease research. However, the focus of our current work is on demonstrating the broad utility of our new scATAC-seq technology and providing novel insights into the epigenetic alterations associated with HGPS, which is beyond what scRNA-seq can reveal. We agree that although exploring the expression patterns in HGPS is beyond the scope of this study, such analysis would be a valuable direction for future research.

REVIEWER #2*****

While I appreciate the detailed explanations provided regarding the ATAC-seq, I still think that, at present, this technology cannot provide functional data without being complemented by RNA-seq. Furthermore, while the authors emphasized that the primary focus of the paper is on describing a new technology, I found that is still focused on progeria.

Additionally, given the inherent variability of epigenetic signatures, the findings derived from a single patient, without validation in additional samples, lack sufficient scientific rigor and reliability.

Response: Thank you for your thoughtful comments and critical evaluation. We have removed all HGPS-related data and discussion and refocused the manuscript entirely on the development and validation of IT-scATAC-seq.

4. Figure 3: The authors compared their IT-scATAC-seq with two available PBMC scATAC-seq datasets generated using different methods (WT 10X and WT s3). However, the rationale for selecting only these two controls is unclear, especially given the eight different scATAC-seq protocols mentioned in De Rop et al., Nat Biotech 2023, which the authors cite.

1st round of response: Thank you for your question regarding our selection of control datasets in Figure 3, and I apologise for not being clearer in the explanation of the rationale. In our comparison, we opted to use datasets from the study by De Rop et al., Nat Biotech 2023, which extensively characterised PBMCs using eight different scATAC-seq methods: 10x Genomics scATAC-seq (v1, v1.1, v2, multiome and mitochondrial scATAC (mtscATAC)) as well as Bio-Rad ddSEQ, HyDrop and s3-ATAC. The same PBMC cell source (pooled two healthy samples) was used across eight different methods:

“Cryopreserved human PBMCs from one male donor and one female donor were purchased from AllCells and distributed across institutes to generate the following samples.....”.

The eight methods in the report by De Rop et al. are primarily categorised into two groups: droplet-based (10X v1, v1.1, v2, ddSEQ, HyDrop) and sci-based protocols (s3) (10X multiome and mtscATAC are multi-omics profiling, and mitochondrial profiling is a variant of 10X that profiles other modalities rather than chromatin accessibility). We selected one representative dataset from each category—10X v1.1 for droplet-based and s3 for sci-

based—to provide a broad perspective on the most prevalent technologies in the field. By incorporating these datasets, we aimed to minimise discrepancies arising from both distinct technological approaches and potential biological variability. In other words, using one dataset from our own research as a biological WT reference, alongside the two additional WT datasets from De Rop et al. processed by different technologies, ensures our comparison between HGPS and WT PBMCs is robust, minimising the impact of both technological and biological variations.

REVIEWER #2*****

This explanation could be incorporated into the Methods section to assist the reader.

Response: Thank you for your comments. We have added this to the revised **Methods** section.

5. Figure 4: The interpretation of the differences between the controls and the single HGPS sample is challenging to follow. Initially, the authors mention an overall decrease in chromatin accessibility but then shift focus to upregulated peaks. However, several downregulated peaks are also linked to T lymphocyte activation, such as the mTOR-signalling pathway. How do the authors justify this discrepancy? They wrote: “Functional analysis of down-regulated peak-associated genes revealed enrichment in immune response pathways, including cell adhesion, lymphocyte differentiation, chemokine, signalling, cytokine production, and infection-related pathways”. However, cytokine production, signaling, lymphocyte differentiation are terms characterizing the immune system activation. Notably, one of the KEGG terms associated with the downregulated peaks is “cellular senescence”. They concluded: “cellular senescence pathways were significantly enriched across all cell types aligning with accelerated immunosenescence in HGPS observed in previous studies”, however, this conclusion appears to be an overinterpretation based on the GO analysis in Figure 4b.

1st round of response: We thank you for your detailed comments and would like to clarify several aspects regarding our analysis and interpretation of chromatin accessibility changes in HGPS PBMCs.

Our analysis identified the observed overall decrease in chromatin accessibility across all immune cell populations analysed in HGPS (Figure 5a). Contrary to “shift focus to upregulated peaks”, our study examines both regions with increased accessibility (up-regulated peaks) and regions with decreased accessibility (down-regulated peaks) (Figure 5b-c, Extended Data Fig.8), presenting a balanced view of chromatin state alterations in HGPS PBMCs.

For regions with decreased chromatin accessibility (down-regulated peaks, Figure 5c), we conducted an enrichment analysis to identify the pathways and processes potentially affected by LMNA mutation in HGPS patient’s PBMCs. Genes associated with these downregulated peaks (2.5kb \pm of TSS region) were found to be enriched in pathways related to cell adhesion, lymphocyte differentiation, chemokine signalling, cytokine production, and infection responses. This does not indicate or imply immune activation in these populations. Rather, it suggests that genes (could be pro- or anti-cell adhesion, lymphocyte differentiation, infection, cytokine production) that are involved in these immune-related processes exhibited reduced accessibility, potentially affecting the regulation of these pathways. Notably, previous studies on lamin A/C-related immune cell function support this interpretation, showing that lamin A/C mutations or deficiencies hasmpact hasmmune regulation, leading to altered hasmmune signaling, activation, and aging-related immune deficits, which has been reviewed by Saez et al. (2020, PMID: 32854281).

Regarding the KEGG “Cellular Senescence” pathway (has04218), we observed that this pathway was enriched among down-regulated peaks across all analyzed cell populations. It is important to note that has04218 includes both pro-senescent and anti-senescent genes (<https://www.genome.jp/pathway/has04218>). Thus, our analysis only shows that the accessibility (expression regulation) of cellular senescence-related genes is reduced in HGPS PBMCs, which are implicated in HGPS-linked aging phenotypes. This finding does not imply that “cellular senescence” itself is downregulated in these cells, nor does it suggest a causal relationship between decreased accessibility of these genes and senescence phenotypes. We believe this provides valuable insight into the chromatin state changes associated with accelerated aging in HGPS without overinterpreting the data.

Together, our findings reveal a pattern of chromatin accessibility changes in HGPS PBMCs that aligns with documented immunosenescence features, including reduced accessibility in genes linked to immune response pathways and cellular senescence. To avoid inferring any functional implications of these analyses, we have also revised the description of the relevant parts in the manuscript to emphasise the findings that were made for the epigenetic changes.

We hope this clarifies the confusion and addresses your concerns about the data interpretation.

REVIEWER #2*****

This analysis has not been sufficiently improved in the revised version. As the data is derived from lymphocytes, it is unsurprising that most terms pertain to the immune system.

However, unless the authors examine single genes to justify the presence of the same or similar terms in both upregulated and downregulated categories, the conclusions will continue to appear speculative. For instance, while mTOR signaling is detected in both upregulated and downregulated peaks in CD8 cells, the authors only discuss its involvement in the upregulated peaks.

In the GO Biological Process (BP) analysis, the authors stated for upregulated peaks:

“Similarly, GO analysis for biological processes (BP) showed gene activity enrichment in lymphocyte differentiation, immune response-activating signalling pathways, and cell adhesion (Fig. 5b)” However, the same categories are also present in the downregulated peaks. This overlap is misleading and should be clearly addressed in the text.

On the other hand, the term "cellular senescence" appears only in downregulated peaks.

Regarding this, the authors wrote: “Importantly, cellular senescence pathways were significantly enriched across all cell types aligning with accelerated immunosenescence in HGPS observed in previous studies (Fig. 5c)” citing the Gonzalo paper (Cell Reports 2018). However, this cited study, which analyzed experiments conducted on four distinct HGPS-derived fibroblast cell lines, reported an upregulation of immune gene expression specifically in HGPS (measured at the RNA level) and highlighted differences at the single-gene level in the GO category. While the overarching conclusion of that paper links immune-related pathway upregulation to cellular senescence, it did not report the "cellular senescence" among GO terms.

The authors should analyze the terms within the GO class and provide single-gene level tracks to improve the paper’s quality and reliability. Given that cellular senescence is observed in the HGPS downregulated peaks, I would expect preferential inclusion of gero-protective genes.

Again, with only a single replicate, it becomes challenging to identify reliable pathways through intersection and selection. This limitation significantly impacts the robustness of

the findings.

Response: Thank you for your comments. We have removed all HGPS PBMCs-related data and discussion and focused on the development and validation of IT-scATAC-seq.

6. Figure 4: Are the upregulated ATAC-seq peaks in HGPS-derived PBMC indicative of a specific epigenetic chromatin state, or are they also linked to active transcription?

Expression analysis is essential for any interpretation of function and dysfunction.

1st round of response: We appreciate your interest in the relationship between ATAC-seq peaks and active transcription.

Upregulated ATAC-seq peaks reflect regions of increased chromatin accessibility at regulatory elements such as TSS, enhancers, and promoters, which suggest the potential for transcriptional regulation but do not directly measure gene expression.

Importantly, ATAC-seq and scRNA-seq capture distinct regulatory and expression-level information, each providing unique insights into cellular states. In this study, the primary aim is to use IT-scATAC-seq to explore epigenetic chromatin states in HGPS-derived PBMCs, focusing specifically on chromatin accessibility as an indicator of regulatory potential rather than gene expression.

To gain a complete understanding of cellular function and dysfunction(which we neither covered nor intended to cover in this study), expression analysis alone is often insufficient. scRNA-seq, for example, is limited by its reliance on polyadenylated RNA capture, which restricts transcriptomic coverage and omits certain RNA species. Furthermore, gene expression levels do not reliably predict protein abundance, as post-transcriptional modifications and other regulatory mechanisms play crucial roles in shaping cellular functions.

We recognise the value of a multimodal approach to achieve a fuller understanding of chromatin state and transcriptional activity and plan to explore this in future studies as techniques evolve further.

REVIEWER #2*****

This response does not address my question. I inquired whether the intersection of upregulated peaks with specific histone marks reveals a preferential chromatin state

affected by the disease. For instance, are bivalent promoters marked by H3K4me3/H3K27me3 particularly affected in HGPS, as previously reported (Della Valle et al., Sci Transl Med, 2022; Sebestyen et al., Nat Commun, 2020; Salvarani et al., Nat Commun, 2019; Briand et al., Hum Mol Genet, 2018)?

Response: Thank you for your comments. We have removed all HGPS PBMCs-related data and discussion and focused on the development and validation of IT-scATAC-seq.

7. Figure 5: The authors chose to intersect the differentially accessible peaks with LAD mapped in HeLa and HAP-1 cells. While most of the LADs are conserved, they opted for a cancer cell line, which likely has an altered chromatin structure, while HAP-1 is a haploid cell line. Why didn't they consider using LADs from a human primary cell line, such as fibroblasts? In the paper cited by the authors, Kohler et al. (Genome Med. 2020) present LAD data for both wild-type and HGPS fibroblasts. Why did the authors not consider the HGPS LADs instead of those from tumor cells? Additionally, Kohler et al. also provide ATAC-seq data; how are the pathological euchromatin changes conserved between fibroblasts and PBMCs? Are they conserved or cell specific?

1st round of response: We thank you for your question regarding LAD dataset selection. LAD data from primary fibroblasts are indeed scarce due to the limited availability of high-quality lamin A/C ChIP-seq data from these cells. Yet, LAD regions are known to be relatively conserved across cell types Kind et al. Cell. 2015 (PMID: 26365489). It is worth noting that in the cited study by Kohler et al. (Genome Med. 2020), they did not present LAD data for wildtype nor HGPS fibroblast; instead, they also relied on LAD data from HeLa cells rather than directly profiling LADs in fibroblasts, due to the high conservation of LAD domains across cell types.

“For the comparison of LAD- and “solo-WCGW” CpG probe methylation levels between HGPS and control samples, we used previously published locations of lamin A LADs (PMID: 25602132), lamin B LADs (PMID: 18463634) ...”

Following what has been used by Kohler et al., we used the same HeLa-derived LAD dataset for A-LADs. For B-LADs, we used the dataset generated from K562 cells (PMID: 32893442).

In our analysis, we observed cell-type-specific changes in chromatin accessibility across

different immune cell types within HGPS-derived PBMCs, revealing euchromatin alteration signatures unique to each subtype. As for comparison with fibroblast data, Köhler et al. reported relatively fewer differentially accessible peaks in HGPS fibroblasts than we have identified in PBMCs. It is difficult to determine whether these differences are due to cell type or variations in analytical methods. Interestingly, FOS2 and BACH2 (among other AP1 family TFs), enriched in differentially accessible regions in HGPS fibroblasts identified in Köhler et al. study, were also found enriched in CD4 Memory cells of in the HGPS patient (Supplementary Figure 9b).

REVIEWER #2*****

The authors wrote in their rebuttal: “LAD data from primary fibroblasts are indeed scarce due to the limited availability of high-quality lamin A/C ChIP-seq data from these cells”. LAD data from primary fibroblasts are not scarce. Lamin-associated domains (LADs) have been mapped in human fibroblasts in several studies, including McCord et al. (Genome Research, 2013), Sadaie et al. (Genes & Development, 2013), Dou et al. (Nature, 2015), and Paulsen et al. (Genome Biology, 2017). For example, in the McCord paper, the authors used two different anti-lamin A/C antibodies (MAB3211 and N18) to map LADs in two biological replicates of HGPS and normal fibroblast cells. Additionally, LAD datasets are available for Jurkat T cells (Robson et al., Genome Research, 2017), which are more similar to the model system used in the present study.

The authors in their rebuttal have incorrectly cited Kind et al. (Genome Research, 2012) as the source for the conservation theory of LADs across cell types. This finding should instead be attributed to Meuleman et al. (Genome Research, 2012). Kind et al. (same lab, but a later study) performed single-cell analyses, reporting significant variability in LAD contacts even within the same cell population, with LADs covering the 35% of the genome and only 15% of LADs classified as structural and present in all cells. This distinction underscores the current understanding of LADs: there are constitutive LADs (cLADs), which are cell-type invariant, and facultative LADs (fLADs), which are cell-type specific and play crucial roles in cell identity. This concept is summarized by Bas van Steensel in his review on Cell (2017), where he estimated that fLADs constitute at least half of all detected LADs.

Another important information is that LAD remodeling is a hallmark of differentiation and disease, as previously shown by Reddy et al. (Nature, 2008) and extensively characterized by

the Collas lab (Lund et al., Genome Research, 2013; Ronningen et al., Genome Research, 2015; Madsen-Osterbye et al., Genome Biology, 2022; Benarroch et al., Cells, 2023). Cancer cells, too, exhibit LAD remodeling, as demonstrated by Lenain et al. (Genome Research, 2017) and Ji et al. (Biochim Biophys Acta Gene Regul Mech., 2020). Thus, due to the extensive alterations in LAD organization seen in cancer cells, they are not suitable as a reference dataset for LAD analysis in healthy cells.

Response: Thank you for your comments. We have removed all HGPS PBMCs-related data and discussion and focused on the development and validation of IT-scATAC-seq.

8. Figure 5: the authors stated Page 10 line 274: “Notably, most immune cell types showed a marked loss of peaks in both A-LADs and B-LADs, but CD4 naive T cells exhibited more gained A-LADs-associated peaks (393 gained vs. 297 lost) and comparable alterations in B-LADs-associated peaks (1,146 gained vs. 1,194 lost), suggesting cell-type-specific resilience to HGPS-induced chromatin alterations”, however, this is an overinterpretation given the lack of Lamin CHIP-seq data in these cells.

1st round of response: We thank you for your comments and acknowledge that Lamin A/C ChIP-seq data would provide a more direct assessment of LAD structural changes in HGPS. However, our study’s objective is to investigate chromatin accessibility changes in specific cell types in HGPS PBMCs, with LAD regions (which are conserved among human cell types) of particular interest, rather than to examine broader structural changes in LAD domains themselves.

Our statement regarding “resilience” in CD4 naive T cells reflects the observation that, unlike other immune cell types, these cells exhibited a unique pattern of peak gains in A-LADs (393 gained vs. 297 lost). This pattern suggests a distinct chromatin accessibility change pattern within the LADs region of CD4 naive T cells under HGPS conditions rather than implying structural resilience in LAD regions. We have tone-downed the text to clarify that this interpretation is based on observed chromatin accessibility patterns within LAD-associated regions relative to other types of cells and does not imply direct structural changes in LADs of patient’s cell populations without ChIP-seq confirmation.

REVIEWER #2*****

I believe there is still an element of overspeculation in this statement. When you assert that “...CD4 naive T cells exhibited significantly more gained A-LADs-associated peaks...” it gives the impression that these peaks are located within LADs. However, without Lamin ChIP-seq data for these cells, such a claim cannot be supported. On the other hand, this statement does not contribute meaningfully to the paper’s core message and, in my opinion, can be omitted.

Response: Thank you for your comments. We have removed all HGPS PBMCs-related data and discussion and focused on the development and validation of IT-scATAC-seq.

9. Figure 6: How did the authors select the set of genes shown in Figure 6a? Are all the genes with reduced chromatin accessibility in LAD across various cell populations included, or did they choose only the most interesting ones?

1st round of response: We thank you for the question regarding the selection criteria for the set of genes shown in Figure 6a. The genes depicted are representative of regions with altered chromatin accessibility in LADs across various cell populations, specifically those that overlap with findings from a previously published aging-related study. As outlined in the manuscript, we focused on genes that exhibited significant loss of chromatin accessibility in LADs ($p < 0.05$) and correlated with decreased expression associated with chronological aging. This targeted approach ensured that we included genes with known relevance to both HGPS and aging.

To clarify, the complete list of overlapped genes with significantly altered accessibility is detailed in Supplementary Data 5 and 6. Among them, we selected a subset that best represented the key findings of our study.

The above has also been noted in the original manuscript: “To identify regions in HGPS PBMCs that are differentially regulated and associated with aging, we compared chromatin accessibility changes in HGPS samples with genes known to be affected by chronological aging. We focused on genes differentially expressed with chronological age with a negative Z-score, indicating decreased expression with aging, and mapped these regions where LAD-associated chromatin accessibility was significantly lost in HGPS compared to healthy controls”.

REVIEWER #2*****

In their response, the authors kindly copied and pasted the same explanation from the text. I apologize if my initial request was not sufficiently clear. When selecting genes, the authors should specify the criteria employed to avoid the appearance of cherry-picking. The authors wrote: “We focused on genes differentially expressed with chronological age with a negative Z-score, indicating decreased expression with aging, and mapped these regions where LAD-associated chromatin accessibility was significantly lost ($p < 0.05$) in HGPS compared to healthy controls. A set of genes within A-LADs and B-LADs, that showed reduced chromatin accessibility in HGPS across various immune cell types were pinpointed (Fig. 7a and Supplementary Table 5-6)”.

However, I could not find information regarding the dataset and cell types used for the differential gene expression analysis. Additionally, they should report the percentage of overlap between aging-associated dysfunctional genes and the upregulated and downregulated peaks.

Response: Thank you for your comments. We have removed all HGPS PBMCs-related data and discussion and focused on the development and validation of IT-scATAC-seq.

10. Figure 6: With a set of interesting genes, it would be beneficial to include their expression profiles, even in a bulk population. RNA-seq data for HGPS patients is available, although not specifically for PBMCs.

1st round of response: We thank the reviewer for suggesting the inclusion of gene expression profiles. However, this study primarily focuses on cell type-specific chromatin accessibility changes in PBMCs from an HGPS patient, assessed through single-cell epigenetic analysis rather than gene expression levels. While RNA-seq data exists for HGPS fibroblasts, it lacks the cell-type specificity necessary for our examination of distinct immune subtypes within PBMCs. For instance, we observed reduced accessibility of AK5, specifically in CD8 Memory cells, whereas genes like ABLIM1 showed consistent changes across various immune cell types. The expression status of AK5 in fibroblasts would not necessarily relate to its regulation in CD8 Memory cells. These findings emphasise the importance of analysing

chromatin accessibility at a cell-type-specific level. Given that fibroblast expression data does not capture the complex regulatory landscape of PBMCs, it would not effectively support the immune cell-specific observations of this study.

REVIEWER #2*****

I agree, and for this reason, I believe the paper should refrain from speculating on specific genes altered in HGPS, particularly when highlighting a single gene based on data from a single patient.

Response: Thank you for your comments. We have removed all HGPS PBMCs-related data and discussion and focused on the development and validation of IT-scATAC-seq.

11. Figure 6: PRMT6 is located within a Lamin B domain, whereas HGPS results from a mutation in Lamin A. Is the gene situated within a domain shared by both Lamin A and Lamin B? If not, how do the authors comment this detail?

1st round of response: PRMT6 is located exclusively within B-LADs (Lamin B-associated LADs) and not within A-LADs (Lamin A-associated LADs) in the reference datasets we used. Its reduced chromatin accessibility was observed specifically in the promoter regions of CD4 Memory and CD8 Naive T cells. We focused on LAD-associated regions as they represent areas in close contact with the nuclear lamina, making them critical for understanding potential regulatory disruptions linked to nuclear architecture.

Although it remains uncertain how A-LADs and B-LADs are differentially affected in HGPS-derived PBMCs and how these changes influence shifts in chromatin accessibility, our findings suggest that

HGPS mutations lead to widespread changes across both A-LAD and B-LAD-associated domains, as well as in predominantly in non-LAD regions (Figure 7c). These observations align with findings by Köhler et al. (Genome Medicine, 2020) in HGPS fibroblasts, where significant chromatin alterations were noted within LADs and largely extended to broader genomic areas. Thus, while PRMT6's reduced accessibility in B-LADs serves as a representative example of specific regulatory changes, the overall nuclear lamina dysfunction in HGPS likely drives complex, genome-wide chromatin remodeling that goes beyond the immediate associations with lamin A/C or lamin B.

REVIEWER #2*****

These observations should also be included in the text to tone down the interpretation of the functional relevance of these findings.

Response: Thank you for your comments. We have removed all HGPS PBMCs-related data and discussion and focused on the development and validation of IT-scATAC-seq.

12. Figure 6: The siRNA data is unconvincing to address the role of PRMT6 in premature senescence. As expected, reduced PRMT6 determines cellular senescence as previously described, but its expression is crucial in the HGPS? Why didn't the authors attempt to overexpress PRMT6 in the MSCs differentiated from HGPS patient-derived iPSCs to revert the premature senescence?

1st round of response: We appreciate the reviewer's suggestion to include PRMT6 overexpression experiments. However, the primary aim of this study is to demonstrate the application of IT-scATAC-seq and validate the identification of genes with reduced chromatin accessibility in HGPS-derived PBMCs. We focused on knockdown experiments to investigate the role of PRMT6 in cellular senescence and chromatin state changes, aligning with our methodology and findings.

While we agree that overexpression studies could offer additional insights into reversing the senescence phenotype, such experiments are beyond the current scope, which primarily aims to confirm the downregulation of PRMT6 in the context of stem cell aging. We have also taken care to avoid overinterpreting our results regarding PRMT6's potential to prevent HGPS-associated aging. Future research may explore whether PRMT6 overexpression could rescue HGPS phenotypes in iPSC-derived models, extending the findings reported here.

REVIEWER #2*****

These data should be removed

Response: Thank you for your comments. We have removed these data accordingly.

Reviewer #3 (Remarks to the Author):

Here Wei Jin et al. present IT-scATAC-seq, a new approach for single-cell ATAC-seq. Their method is based on indexed Tn5 tagmentation followed by pooling and distribution into wells for barcoded PCR (a 2-step combinatorial indexing method). They show some benchmarking analysis to compare with other established methods such as 10x, Fluidigm, and sci-ATAC-seq and demonstrate some higher quality metrics. Importantly, the reported cost for IT-scATAC-seq is only around \$0.01/cell, lower than 10x genomics and some other methods. However, this is still higher than the reported cost for EasySciATAC. Overall, the method itself seems to be a sound approach capable of generating high-quality data across a range of sample types.

This work also goes into some detail about the biology of HGPS. To me, this seems to distract from the main message of the paper, the presentation of a novel technology. It is certainly an advantage to show that this technology can work on clinical samples, but the major focus should remain on the evaluation of IT-scATAC-seq as a new technology. More emphasis can be given to the comparison with existing technologies. For example, the authors should discuss limitations of the IT-scATAC technology (the need for indexed Tn5, lower throughput and higher cost relative to sci-ATAC-seq methods (including EasySciATAC), and the need for FANS). Extended data figure 6 can be moved to the main figure panels to provide the details regarding cost, as lower cost is emphasized as a major advantage of the method. Some of the biological results could be shifted to the supplement. Regarding cost, the cost of Tn5 is currently not included in the calculation.

Overall, this appears to be a sound method for scATAC-seq capable of generating high quality data. Some non-trivial setup is required if labs are wishing to adopt this, most significantly the production of Tn5 and use of FANS, and the cost is not as low as the 3-step combinatorial approaches or the droplet combinatorial indexing approaches (some of which are not mentioned in the paper). In my opinion the in-depth discussion of HGPS only serves to distract from the main message of this paper and should be significantly scaled back in its presentation.

Response: We sincerely appreciate your thoughtful evaluation of our manuscript and your recognition of IT-scATAC-seq as a robust methodological innovation. We fully agree that the primary focus should remain on the technical merits, benchmarking, and comparative evaluation of the method, and we have revised the manuscript accordingly to align with this priority.

To address your feedback, we have made the following revisions.

a) Regarding the biology of HGPS, we have **removed all HGPS-related data and discussions** from the manuscript. The focus is now only on the development, validation, and benchmarking of IT-scATAC-seq, emphasising its technical performance across diverse sample types. We have added a detailed step-by-step protocol as the **Supplementary Method** to allow more labs to adopt this.

b) To emphasise comparisons with existing methods, the multiplexing strategy, machinery dependency, throughput and cost, etc., is now summarised in **Supplementary Table 3** (Also shown below).

Supplementary Data 2. Comparisons of scATAC-seq methods.

scATAC-seq methods	Tn5	Sequencing	Equipment	Indexing strategy	Collision/error rate	Manual labour	Throughput	Library preparation cost per cell(\$)
sci-ATAC-seq	Custom-made, ≥96 Tn5	Custom-made Read 1 and 2, Whole lane sequencing	Flow cytometry	Tn5 + indexed PCR	Sorting error + Barcode collision	Heavy	10 ⁴	>1.00
Plate-based scATAC	Nextera	Nextera, Whole lane sequencing	Flow cytometry	indexed PCR	Sorting error	Heavy	10 ² -10 ³	0.18
Fluidigm C	Nextera	Nextera, Whole lane sequencing	Microfluidics	indexed PCR	Distribution error	Light	10 ²	>1.00
μATAC-seq	Nextera	Nextera, Whole lane sequencing	Takara iCELL8	indexed PCR	Barcode collision	Light	10 ³	0.81
10x Chromium	Nextera	Nextera, Whole lane sequencing	10x Chromium	indexed PCR	Barcode collision	Light	10 ⁴	0.50-1.00
Bio-Rad dsciATAC	Custom-made, ≥96 Tn5	Custom-made Read 1 and 2, Whole flow sequencing	Single-Cell Isolator	Tn5 + indexed	Barcode collision	Heavy	10 ⁵	0.05-0.10
EasySciATAC	Custom-made, 384 Tn5	Custom-made Read 1 and 2, Whole flow sequencing	No	Tn5 + Ligation + indexed PCR	Barcode collision	Heavy	10 ⁶	≤0.006
IT-scATAC-seq	Custom-made, 10-24 Tn5	TruSeq, Multiplexing with other libraries	Flow cytometry	Tn5 + Two indexed PCR	Sorting error	Light	10 ⁴ -10 ⁵	≤0.01

c) As you recommended, we have **expanded the Discussion** to address key limitations of IT-scATAC-seq:

“While IT-scATAC-seq offers several advantages, it also has some trade-offs, primarily due to its in-house nature. First, indexed Tn5 transposase may be a barrier for labs without enzyme preparation capabilities, though commercial Tn5 is available. Second, although IT-scATAC-seq simplifies the workflow, its cell throughput—given equivalent time and labour—is lower than sci-ATAC-seq⁷ and its derivatives like EasySciATAC²⁰. Third, IT-scATAC-seq improves resolution and lowers barcode misassignment through FANS but requires flow cytometry resources and constitutes the most time-consuming stage of the workflow. Future optimisations could develop alternative nuclei-handling strategies to reduce FANS dependency and improve throughput.”

Considering the cost and throughput compared with EasySciATAC, we fully agree that unparalleled cost efficiency and scalability for projects profiling millions of cells. Its three-step combinatorial indexing and robust data integration make it an invaluable tool for constructing large, comprehensive cell atlases.

In contrast, IT-scATAC-seq is optimised for smaller-scale studies (10,000–100,000 cells), focusing on workflow simplicity and rapid implementation. By reducing the protocol to two indexing steps, IT-scATAC-seq provides a more accessible and streamlined workflow, particularly suitable for labs conducting hypothesis-driven research or working in resource-limited settings.

While the \$0.01 per-cell cost of IT-scATAC-seq is based on calculation for profiling 10,000 cells, the cost decreases with larger datasets due to the scalable nature of the indexing process (which primarily involves assembling additional index combinations). For experiments involving up to one million cells, the per-cell cost would be further reduced, though the sorting time via FANS would proportionally increase. Nonetheless, IT-scATAC-seq remains highly cost-effective for focused, mid-scale studies with prioritised ease of use, flexibility, and rapid turnaround. By complementing methods like EasySci, IT-scATAC-seq broadens access to high-quality single-cell epigenomic tools for a wider range of research applications.

By revising the manuscript to prioritise rigorous technical validation and benchmarking, we emphasise IT-scATAC-seq as a versatile and high-performance alternative to current single-cell omics methods. We deeply appreciate your insightful feedback, which has significantly strengthened the clarity and scientific impact of this work.

REVIEWERS' COMMENTS

Reviewer #2 (Remarks to the Author):

authors addressed my concerns

Response: We appreciate your valuable feedback and are glad to hear that your concerns have been addressed. Thank you for your constructive comments.

Reviewer #3 (Remarks to the Author):

The authors have addressed most of my comments and the revised version presents a much more focused evaluation of the new scATAC-seq technology. I still recommend moving supplementary figure 6 into the main figure panels (this suggestion was not addressed in the response). Supplementary Table 2 serves as a nice summary comparison of the different methods and will be a good resource for labs choosing a method to suit their needs. However, the presentation of the “Sequencing” column requires some revision. EasySciATAC does not require custom sequencing primers. EasySciATAC and 10x Genomics do not require whole lane sequencing (can multiplex with other libraries).

Response: We appreciate your thorough review and constructive feedback, which have helped improve the clarity and focus of our manuscript.

Regarding Supplementary Figure 6, we acknowledge your suggestion to include it in the main figures. Given the space constraints and the need to maintain a balanced presentation of all key aspects, we have opted to retain it in the supplementary materials. To ensure its relevance is clear, we have revised the text accordingly to better highlight its content within the main manuscript.

We also appreciate your careful review of Supplementary Table 1 and your suggestions for improving the sequencing column. We have now revised this section to accurately reflect that EasySciATAC does not require custom sequencing primers and that both EasySciATAC and 10x Genomics can be multiplexed with other libraries. We appreciate the opportunity to refine this comparison and ensure it provides a clearer resource for researchers choosing among different methods.

Thank you again for your constructive comments and for your efforts in reviewing our work.